# The Kinetics and Stoichiometry of Metal Cation Reduction on Multi-Crystalline Silicon in a Dilute Hydrofluoric Acid Matrix

**DOI:** 10.3390/nano10122545

**Published:** 2020-12-17

**Authors:** Stefan Schönekerl, Jörg Acker

**Affiliations:** Department of Physical Chemistry, Faculty of Environment and Natural Sciences, Brandenburg University of Technology Cottbus-Senftenberg, Universitätsplatz 1, 01968 Senftenberg, Germany; joerg.acker@b-tu.de

**Keywords:** kinetics, stoichiometry, metal deposition, silicon, hydrofluoric acid, course of reaction

## Abstract

In this study, the process of metal cation reduction on multi-crystalline silicon in a dilute hydrofluoric acid (HF) matrix is described using Ag(I), Cu(II), Au(III) and Pt(IV). The experimental basis utilized batch tests with various solutions of different metal cation and HF concentrations and multi-crystalline silicon wafers. The metal deposition kinetics and the stoichiometry of metal deposition and silicon dissolution were calculated by means of consecutive sampling and analysis of the solutions. Several reaction mechanisms and reaction steps of the process were discussed by overlaying the results with theoretical considerations. It was deduced that the metal deposition was fastest if the holes formed during metal ion reduction could be transferred to the valence bands of the bulk and surface silicon with hydrogen termination. By contrast, the kinetics were lowest when the redox levels of the metal ion/metal half-cells were weak and the equilibrium potential of the H_3_O^+^/H_2_ half-cells was high. Further minima were identified at the thresholds where H_3_O^+^ reduction was inhibited, the valence transfer via valence band mechanism was limited by a Schottky barrier and the dissolution of oxidized silicon was restricted by the activity of the HF species F^−^, HF_2_^−^ and H_2_F_3_^−^. The findings of the stoichiometric conditions provided further indications of the involvement of H_3_O^+^ and H_2_O as oxidizing agents in addition to metal ions, and the hydrogen of the surface silicon termination as a reducing agent in addition to the silicon. The H_3_O^+^ reduction is the predominant process in dilute metal ion solutions unless it is disabled due to the metal-dependent equilibrium potential of the H_3_O^+^/H_2_ half-cell and the energetic level of the valence bands of the silicon. As silicon is not oxidized up to the oxidation state +IV by the reduction of the metal ions and H_3_O^+^, water is suspected of acting as a secondary oxidant. The stoichiometric ratios increased up to a maximum with higher molalities of the metal ions, in the manner of a sigmoidal function. If, owing to the redox level of the metal half-cells and the energetic level of the valence band at the metal–silicon contact, the surface silicon can be oxidized, the hydrogen of the termination is the further reducing agent.

## 1. Introduction

The wet chemical structuring of silicon is of great importance for various applications in photonics, microsystems technology, microelectronics and sensor technology [1]. The process of metal-assisted etching to create special spatial structures is the topic of interest [2,3,4]. Despite very intensive investigations, not all details of the reactions occurring during metal-assisted etching have yet been clarified [5]. An essential partial aspect of the research question refers to the process of electroless precious metal cation reduction on silicon in a diluted hydrofluoric acid (HF) matrix.

At the beginning of this process, the metal nucleation starts mainly at crystal defects or kinks on the silicon surface [6,7,8]. Further metal growth is a highly dynamic process in which already formed metallic nuclei are oxidized again to metal cations in order to subsequently be deposited again as nuclei at energetically more favorable locations [9,10]. The diffusion of the metal cations to the silicon surface and further to the nuclei is the rate-limiting factor of metal deposition [11,12,13,14,15]. The nuclei expand to nanoclusters [16], islands or dendrites [17,18,19,20] on the silicon surface according to the Stranski–Krastanov and Volmer–Weber principles [21,22], and then form thin (nm to µm) [6,12,19,23,24], usually microporous layers [16,17] on the silicon surface. Adhesion forces between metal and silicon are usually low [6,25,26,27]. Depending on the metal, different shapes are formed on the silicon surface, such as pits (Cu) [18,28,29], pores (Ag, Au, Pt) [6,16,19,30], threads or nanowires (Ag, Au) [16,18,19,23,24,29,30,31,32,33,34]. The silicon removal occurs linearly in time, independent of the type of growth of the metals, the type of silicon doping, the amount of doping or the orientation of the silicon surface [10,19,35]. The metal deposition is also linear up to a certain thickness of the metal layer [36] and then follows first-order kinetics [12,36,37].

The reactions behind this process are reported in different ways in the literature. According to a prominent theory, there is a kinetically controlled transfer of two electronic holes h_v_^+^ from the metal cation/metal redox pair Me^z+^/Me to the valence band of the silicon (divalent mechanism, Equation (1)) [16,38,39,40], or in the case of copper deposition, an electron uptake of the metal ions via the conduction band mechanism of silicon [11]. Furthermore, a twofold electron transfer e_c_^−^ from the silicon conduction band to the hydrogen ions or other HF species contained in the solution results in the formation of Si^4+^ and molecular hydrogen (Equation (2)) [16,38,39,40].
(1)$$\frac{2}{z}{\;\mathrm{Me}}^{z+}\to \frac{2}{z}\;\mathrm{Me}\;+2{\;h}_{v}{}^{+}\;\mathrm{and}\;\mathrm{Si}\;+\;2{\;h}_{v}{}^{+}\to {\;\mathrm{Si}}^{2+}$$
(2)$${\mathrm{Si}}^{2+}\to {\mathrm{Si}}^{4+}+\;2{\;e}_{c}{}^{-}\;\mathrm{and}\;2{\;H}^{+}+\;2{\;e}_{c}{}^{-}\;\to {\;H}_{2}$$

According to another prominent thesis, there is a fourfold valence transfer between the metal cations and the silicon atoms via the silicon valence band (tetravalent mechanism, Equation (3)) without the formation of molecular hydrogen. It is characterized by a fast reaction with bulk silicon atoms [18,24,29,41] and rather slow kinetics with the silicon atoms of the hydrogen surface terminations [26,41] formed by the HF matrix.
(3)$$\frac{4}{z}{\;\mathrm{Me}}^{z+}\to \frac{4}{z}\;\mathrm{Me}\;+4{\;h}_{v}{}^{+}\;\mathrm{and}\;\mathrm{Si}\;+\;4{\;h}_{v}{}^{+}\to {\;\mathrm{Si}}^{4+}$$

It was shown in the study by daRosa et al. [42] that the stoichiometric ratios of Cu^2+^ reduction to metallic Cu and the associated oxidation and dissolution of silicon during the reaction of Cu^2+^ with silicon in HF matrix varied between 3:4 and 2:1 mol∙kg^−1^:mol∙kg^−1^. This was interpreted as an indication of a change between the divalent and tetravalent reaction mechanism depending on the concentration of Cu^2+^ and HF. In this context, Chartier et al. [43] formulated Equation (4) for the anodic reaction on silicon.
(4)$$\mathrm{Si}\;+\;6\;\mathrm{HF}\;+{n\;h}_{v}{}^{+}\to H_{2}{\mathrm{SiF}}_{6}+{n\;H}^{+}+\left(\frac{4-n}{2}\right)H_{2}$$

The difference in the redox potentials [17,44,45], i.e., free enthalpy of the redox half-cells involved, is considered as the driving force of the reactions [11,17]. After the initial metal nucleation, the consecutive metal cation reduction occurs at the metal–silicon interface. The background is the Fermi energy alignment at the metal–silicon interface and a valence and conduction band bending of the silicon resulting from the difference in the work function for the metal and the silicon. Depending on the difference in work functions, this process causes the transfer of holes from the metal cation reduction at the level of Fermi energy through the higher work function element (usually the metal) to the valence band of the lower work function element (usually the silicon) [12,19,20,29,38,46]. Metals with particularly high work functions, such as platinum, can also develop a Schottky barrier behavior at the metal–silicon contact, which acts as a potential barrier and inhibits valence transfer kinetically [47]. The lower the Schottky barrier height [20,46,48] and the greater the metal valence band overlap with the silicon [29], the faster the holes resulting from metal reduction should be transferred to the silicon [20,23,24,33,46,48].

The roles of HF and its dissociation products are ambivalent in the process. Hydrogen ions hypothetically participate in silicon oxidation, as described in Equations (2) and (4) [16,38,39,40,43,49,50,51,52]. Undissociated HF (HF_undiss._) [13,39,53,54] and/or the anionic dissociation products of the HF (F^−^ [33,36,55,56,57], HF_2_^−^ [43,58,59]) are suspected of being responsible for the dissolution reaction of the oxidized silicon, resulting in the formation of the formal end product SiF_6_^2−^ [53], according to Equation (5).
(5)$${\mathrm{Si}}^{4+}+6{\;F}^{-}\mathrm{or}\;3{\;\mathrm{HF}}_{2}{}^{-}\mathrm{or}\;6{\;\mathrm{HF}}_{\mathrm{undiss}.}\to {\mathrm{SiF}}_{6}{}^{2-}\pm \;3..6{\;H}^{+}$$

The kinetics of the spatial growth of a metal on the silicon and the dissolution of the silicon should be faster with higher HF concentrations [13,50,51]. By contrast, the formation of a dense metal deposit on the silicon surface can restrict the access of HF species to the oxidized silicon, and thus, reduce the rate-limiting process of silicon dissolution [33].

The formation of hydrogen terminations on the silicon in the HF-matrix [60] has the consequence that only those metals can be deposited whose Me^z+^/Me half-cell redox level is higher than that of the hydrogen half-cell [11,56]. Furthermore, it is known that fluoride adsorption on the metal deposited can generate a strong dipole moment in the boundary layer, which increases the work function of the metal [61], thereby influencing the band bending mechanism at the metal–silicon contact, and in turn, the valence transfer between metal cation and silicon.

In conclusion, it can be stated that the process of electroless metal deposition is characterized by contradictory findings and theories. So far, a comparative study of the electroless deposition of different metals on silicon under identical conditions is missing.

The aim of this study was to determine and compare the kinetics of silver, copper, gold and platinum deposition on silicon experimentally, to determine the influences of different concentrations of the metal cations Ag^+^ and Cu^2+^; the anionic complexes AuCl_4_^−^ and PtCl_6_^2−^; and HF on the kinetics of the respective metal depositions and to draw conclusions regarding the enhancing or limiting factors of the processes.

The data obtained experimentally were further evaluated regarding the stoichiometric ratios of metal cation reduction to metal state, silicon oxidation and consecutive dissolution. Conclusions concerning general and metal-dependent reaction processes will be drawn in the summary of the results.

## 2. Materials and Methods

Common procedure: The experiments performed consisted of batch tests with solutions of different compositions of ultrapure water (18 MΩ resistance, PURELAB^®^ flex, ELGA VEOLIA, Celle, Germany); HF (40%, *w*/*w*, p. A. quality, EMSURE^®^, Merck KGaA, Darmstadt, Germany); and Ag^+^, Cu^2+^, AuCl_4_^−^ or PtCl_6_^2−^ in different concentrations. Multi-crystalline silicon wafers were added in all experiments in the same amount. The batches were stirred under argon atmosphere and sampled several times according to a fixed schedule. The dissolved Si and Ag, Cu, Au or Pt species were subsequently determined by inductively coupled plasma atomic emission spectroscopy (ICP OES).

Preparation of the solutions: A stock solution based on AgF was prepared for the silver batch tests. It was obtained by reacting AgNO_3_ (≥99%, ACS reagent, Sigma-Aldrich, St. Louis, MO, USA) with NaOH (25%, *w*/*w*, extra pure, Carl Roth, Karlsruhe, Germany) to form Ag_2_O and its subsequent conversion with 40% HF solution (*w*/*w*), resulting in a final concentration of 2.9 mol∙kg^−1^ AgF/2.9 mol∙kg^−1^ HF. A total of 27 solutions with a respective mass m_sol_ of 0.060 ± 0.001 kg were prepared from the dilution of this stock solution and the addition of 40% HF solution (*w*/*w*). Initial HF molalities of *b HF (diss., t = 0)* = 0.059 ± 0.001, 0.295 ± 0.004 and 1.472 ± 0.072 mol∙kg^−1^ were set for one-third each. The initial Ag^+^ molalities *b Ag (diss., t = 0)* were between 1.2 × 10^−6^ and 1.3 × 10^−1^ mol∙kg^−1^.

The Cu(II) solutions were based on the addition of CuSO_4_·5H_2_O (≥98%, ACS reagent, Sigma-Aldrich, St. Louis, MO, USA). In summary, 24 solutions were prepared with respective mass m_sol_ of 0.061 ± 0.001 kg. The initial Cu(II) molalities *b Cu (diss., t = 0)* were between 2.1 × 10^−5^ and 4.1 × 10^−1^ mol∙kg^−1^. The initial HF molalities were 0.060 ± 0.001 mol∙kg^−1^ in eight batches, 0.299 ± 0.004 mol∙kg^−1^ in nine batches and 1.499 ± 0.004 mol∙kg^−1^ in seven batches.

A commercially available HAuCl_4_ solution (40–44% Au, *w*/*w*, ≥98%, ACS reagent, Sigma-Aldrich, St. Louis, MO, USA) was used as the stock solution for the Au(III) batches. From this, nine solutions with initial AuCl_4_^−^ molalities of *b Au (diss., t = 0)* = 3.3 × 10^−7^ to 9.3 × 10^−2^ mol∙kg^−1^ were prepared. The masses of the solutions (m_sol_) were 0.060 ± 0.001 kg. The initial HF molalities of these solutions were 1.499 ± 0.006 mol∙kg^−1^. Furthermore, four identically concentrated HF solutions with AuCl_4_^−^ molalities between 6.7 × 10^−5^ and 3.7 × 10^−3^ mol∙kg^−1^ were prepared with the addition of 2.2 × 10^−4^ mol∙kg^−1^ of para-Chlorobenzoic acid (≥98%, Alfa Aesar, Kandel, Germany).

The Pt(IV) etching solutions contained additions of H_2_PtCl_6_·6H_2_O (≥37.5% Pt base, ACS reagent, Sigma-Aldrich, St. Louis, MO, USA). Ten batches were prepared with initial masses m_sol_ of 0.060 ± 0.001 kg and initial PtCl_6_^2−^ molalities between *b Pt (diss., t = 0)* = 4.2 × 10^−7^ and 9.7 × 10^−3^ mol∙kg^−1^ at initial HF molalities of 1.493 ± 0.033 mol∙kg^−1^. Analogously to the Au(III) batches, four further HF solutions with initial PtCl_6_^2−^ molalities between 1.6 × 10^−5^ and 1.7 × 10^−3^ mol∙kg^−1^ with the addition of 2.2 × 10^−4^ mol∙kg^−1^ of para-Chlorobenzoic acid were prepared.

Silicon wafers: ten undoped multi-crystalline silicon wafers (15 mm edge length, 0.175 mm thickness, SCHOTT Solar, Mainz, Germany) pretreated with 1% HF solution were used for each experiment, each with a total mass of m_Si_ = 0.92 ± 0.03 g and a total surface area of A_Si_ = 46 cm^2^.

Experimental procedure: Each HF solution was transferred into a 150 mL low-density polyethylene vessel (VITLAB, Großostheim, Germany) and stirred using a polytetrafluoroethylene (PTFE)-coated magnetic stir bar (AlNiCo, 8 × 20 mm, VWR collection, Radnor, PA, USA) and a stirring plate (RT 10 power, IKA^®^, Staufen, Germany) at about 125 rpm at T = 295 ± 2 K. A GL-45 screw cap made of PTFE with three attached GL-14 ports was used to close the vessel (Bohlender, Grünsfeld, Germany). The HF pretreated silicon wafers were attached to one of the ports using a PTFE thread. The vessel was rinsed via the other two ports with an argon flow of 45 ± 1 mL via PTFE hose lines (using a Gas Flow Meter, Yokogawa, Ratingen Germany and Ar 6.0, Praxair, Danbury, CT, USA). The silicon wafers were then transferred into the etching solution by pulling the PTFE thread externally. The stirring of the etching solution and rinsing with argon was maintained continuously. The solutions were sampled seven times in a period of t = 3600 s (Ag, Cu, Au) or t = 14,400 s (Pt) via the free GL14 port for the time of t = 0, 300, 600, 1200, 1800, 2700 and 3600 s (Ag, Cu, Au) or t = 0, 1200, 2400, 4800, 7200, 10,800 and 14,400 s (Pt).

Sample analysis: All samples were diluted in 1% HF matrix (*v*/*v*, dilution of 40% HF solution). They were subsequently analyzed by ICP OES (iCAP Duo 6500, Thermo-Fisher, Waltham, MA, USA). The spectral intensities λ = 243.7, 328.0 and 338.2 nm for the Ag; λ = 224.7, 324.7 and 327.3 nm for Cu; λ = 201.2, 242.7 and 267.5 nm for Au; λ = 214.4, 265.9 and 306.4 nm for Pt; and λ = 212.4, 251.6 and 252.8 nm for Si were measured axially to the torch. The molalities of the elements (+/− confidence interval) b (mol∙kg^−1^) were determined using external calibrations based on dilution series of 1000 mg∙kg^−1^ standard stock solutions (Cu, Ag, Au, Pt and Si, Certipur^®^, Merck KGaA, Darmstadt, Germany).

## 3. Results and Discussion

### 3.1. Kinetics of the Metal Depositions at One Initial Metal Ion and HF Concentration Levels

Figure 1a–d shows the results of the Ag, Cu, Au and Pt deposition experiments with initial metal ion molalities (*b Me (diss., t = 0)* with Me = Ag, Cu, Au or Pt) of ≈ 1 × 10^−2^ and 1.5 mol∙kg^−1^ HF. The four following diagrams depict the molalities of the dissolved Ag, Cu, Au or Pt species at the respective sampling time *t = i* (*b Me (diss., t = i)*) and the respective findings for dissolved Si (*b Si (diss., t = i)*).

The decrease of *b Me (diss., t = i)* can be attributed to the reduction of Ag(I), Cu(II), Au(III) or Pt(IV) to the metal state, forming a metal deposit on the silicon wafers. The increase of dissolved silicon in the solutions is based on the oxidation of the silicon and the consecutive dissolution of the oxidized silicon by HF species.

A sectioned exponential trend of the molalities of the species can be observed for all types of metal depositions. This is in accordance with the behavior described in the literature [12,36,37]. Metal deposition in the first phase starts phenotypically at the edges of the silicon wafers, i.e., around the defects created by the cutting of the wafers. It also occurred successively at the wafer surfaces (nucleation). In the second phase, a metal film grows on the entire silicon surface, which appears sponge-like in the silver deposition due to its dendritic growth but appears compact and flat in the other metals’ deposits.

The mathematical separation of the two phases is the intersection *t*_1/2_ of the two exponential functions determined, shown in Figure 1a–d. The different exponents of the exponential functions of the first and second phase *k*_1_ (s^−1^) and *k*_2_ (s^−1^), further described as metal deposition rates, were calculated according to Equation (6). The parameter *b Me (diss., t*_1/2_*)* mentioned denotes the molality of the dissolved metal ion at the time *t*_1/2_. The value *t*_95%_ was chosen arbitrarily and corresponds to the time at which the initial dissolved metal ion molality was reduced by 95% and *b Me (diss., t*_95%_*)* is the corresponding value (=0.05 ∙ *b Me (diss., t = 0)*).
(6)$$k_{1}=-\ln\left(\frac{b\;Me\;\left(diss.,\;t_{1/2}\right)}{b\;Me\;\left(diss.,\;t\;=\;0\right)}\right)\cdot {\left(t_{1/2}\right)}^{-1}\;\mathrm{and}\;k_{2}=-\ln\left(\frac{b\;Me\;\left(diss.,\;\;t_{95\;\%}\right)}{b\;Me\;\left(diss.,\;t\;=\;0\right)}\right)\cdot \;{\left(t_{95\%}\right)}^{-1}$$

It was found that the metal deposition rates of the first phase *k*_1_ were about a factor of 2 to 3 below the rates of the second phase *k*_2_. This implies that the metal deposits started relatively slowly. The metal deposition accelerated after a mean metal deposition of about 2.0 × 10^−6^ mol per cm² of wafer surface. Subsequently, the amount of metal deposition per time unit decreased constantly, resulting in an exponential decay of the metal cation molality. The duration of the first deposition phase from *t* = 0 to *t*_1/2_ is less than 360 s for the examples of Ag, Cu and Au shown in Figure 1. It is about 10 times longer for platinum deposition. The second phase, defined as the period from *t*_1/2_ to *t*_95%_, is about 5 (Ag, Au, Pt) to 10 times (Cu) longer than the first phase.

The kinetics of the metal deposits over both phases *k*_95%_ were calculated by weighting the metal deposition rates *k*_1_ and *k*_2_ according to the duration of the two phases according to Equation (7).
(7)$$k_{95\%}=k_{1}\cdot \frac{t_{1/2}-t\;=0}{t_{95\%}}+k_{2}\cdot \frac{t_{95\%}-t_{1/2}}{t_{95\%}}$$

Based on this approach, Ag deposition was found to be the fastest (*k*_95%_ ≈ 2.0 × 10^−3^ s^−1^), followed by Au (*k*_95%_ ≈ 1.5 × 10^−3^ s^−1^), Cu (*k*_95%_ ≈ 0.5 × 10^−3^ s^−1^) and Pt (*k*_95%_ ≈ 0.2 × 10^−3^ s^−1^). It was noted that the amount of the silicon dissolution rate *k*_95%_ corresponded to the respective metal deposition rate.

### 3.2. Kinetics of the Metal Depositions at Different Metal Ion and HF Concentration Levels

In Figure 2, the results of Figure 1 were taken (marked by red dots) and supplemented with the results of further batch tests with different initial metal ion and HF molalities. All experiments were performed and evaluated as described in Section 3.1. The results of these experiments are presented as calculated metal deposition rates *k*_95%_ and plotted against the initial metal ion molalities *b Me (diss., t = 0)*, and the initial redox levels of the Me^z+^/Me half-cells *E (Me^z+^/Me) (t = 0)*. The calculation of *E (Me^z+^/Me) (t = 0)* was based on the Nernst equation using the standard potentials of the Me^z+^/Me half-cells [62,63] and the Bromley equations [64] to calculate the activities.

The classification of these results into the overall context reveals that the kinetics of all metal depositions vary considerably. The ranking of the metal deposition kinetics based on the findings in Figure 1 has no general validity. In principle, it can be seen that there is a positive correlation with the initial Ag(I), Cu(II), Au(III) or Pt(IV) molality or the redox level of the Me^z+^/Me half-cell, starting from the lowest selected metal ion molality up to a certain inflection point. This section is denoted as *k*_95%,1_. The functional relationship for silver and copper deposition is logarithmic, and is potential for gold and platinum deposition, as can be derived in the respective equations in Figure 2.

The inflection points identified are located at *b Ag (diss., t = 0)* = 5.5 × 10^−3^ mol∙kg^−1^ or *E (Ag^+^/Ag)* = 0.65 V vs. the standard hydrogen electrode (SHE) and at *b Cu (diss., t = 0)* = 1.0 × 10^−2^ mol∙kg^−1^ or *E (Cu^2+^/Cu)* = 0.27 V vs. SHE. There is also a narrow drop in the kinetics in copper deposition at *b Cu (diss., t = 0)* = 2.5 × 10^−4^ mol∙kg^−1^ or *E (Cu^2+^/Cu)* = 0.23 V vs. SHE. The gold and platinum depositions reveal two inflection points. They are situated at *b Au (diss., t = 0)* = 3.1 × 10^−5^ and 1.5 × 10^−4^ mol∙kg^−1^ or *E (AuCl_4_^−^/Au)* = 0.91 V vs. SHE and 0.92 V vs. SHE, and at *b Pt (diss., t = 0)* = 5.1 × 10^−5^ and 6.0 × 10^−5^ mol∙kg^−1^, which corresponds to a redox potential of *E (PtCl_6_^2−^/Pt)* ≈ 0.68 V vs. SHE.

When the inflection point is exceeded in the silver deposition experimental series, a significant forcing of the kinetics occurs, which is correlated with the initial Ag^+^ molality or the redox level of the Ag^+^/Ag half-cell via a potential function (cf. *k*_95%,2_ in Figure 2a). The maximum silver deposition rate determined was *k*_95%,*max*_ = 6.1 × 10^−3^ s^−1^ at *b Ag (diss., t = 0)* = 1.5 × 10^−1^ mol∙kg^−1^. The results of three experiments within this test series do not follow this trend (cf. gray data points in Figure 2a). For Cu deposition experiments, *k*_95%,*max*_ is 5.3 × 10^−4^ s^−1^ (= result in Figure 1b); for Au, *k*_95%,*max*_ is 1.1 × 10^−2^ s^−1^; and for Pt, *k*_95%,*max*_ is 6.4 × 10^−4^ s^−1^. The metal deposition rate above the inflection point *k*_95%,2_ in the Cu test series correlates negatively, potentially, with the initial Cu^2+^ molality or Cu^2+^/Cu redox level (cf. gray marked points in Figure 2b). The metal deposition rate within the gold deposition series initially drops to a minimum after the first inflection point has been reached.

After exceeding the second inflection point at *b Au (diss., t = 0)* = 1.5 × 10^−4^ mol∙kg^−1^, the metal deposition rate *k*_95%,2_ is positively logarithmically linked with the initial AuCl_4_^−^ molality (cf. in Figure 2c).

The kinetics of platinum deposition are similar to the kinetics of gold deposition up to the second inflection point but at a lower level of *k*_95%,1_. In contrast to gold deposition, *k*_95%,2_ for platinum deposition decreases negatively logarithmically with an increasing initial Pt(IV) molality (cf. in Figure 2d).

### 3.3. Discussion of Metal Deposition Kinetics

The different findings for metal deposition kinetics can be derived by considering the state of the initial redox levels of the Me^z+^/Me half-cells, the valence transfer at the metal–silicon contact and the influence of the HF molality.

Silver: The inflection point of the silver deposition kinetics is an indication of the change of valence transfer between Ag^+^, the hydrogen-terminated Si surface and the bulk silicon, that can be explained by the simplified theoretical energy scheme shown in Figure 3a.

The illustration contains the range of the redox levels of the Ag^+^/Ag half-cells *E (Ag^+^/Ag)* shown in Figure 2a; the work functions of silver *Φ_Ag_* = 4.37 eV ± 0.37 eV [65,66,67,68,69] and hydrogen-terminated silicon *Φ_Si–H_* = 4.36 eV ± 0.19 eV in vacuum [70]; the Fermi energy *E_F_*; and the energetic levels of valence and conduction bands of hydrogen-terminated silicon (*E_v_ (Si–H)* = −0.68 eV and *E_c_ (Si–H)* = 0.65 eV) [71]. The positions of the valence and conduction bands of bulk silicon (*E_v_ (Si_bulk_)* = −0.41 eV and *E_c_ (Si_bulk_)* = 0.71 eV) were calculated based on the difference of the work function of silicon *Φ_Si_* = 4.76 eV ± 0.34 eV [72], the electron affinity of silicon *χ* = 4.05 ± 0.03 eV [73] and the band gap of 1.12 eV [73].

The work functions of Ag *Φ_Ag_* and hydrogen-terminated silicon *Φ_Si–H_* are, on average, almost identical under a vacuum. When Ag is deposited on Si and the Fermi energy *E_F_* at the silver–silicon contact is equalized, there is no significant band bending of the valence and conduction band of the silicon at the silver–silicon contact due to the nearly identical work functions (Δ*E* = *Φ_Ag_* − *Φ_Si–H_* = 4.37 eV − 4.36 eV = 0.01 eV).

If it is hypothetically assumed that the difference in the work function of Ag and hydrogen-terminated silicon Δ*E* = 0.01 eV has an equal effect on the band bending of bulk silicon and hydrogen-terminated silicon, the energy level of the valence band of bulk silicon at the silver–silicon contact *E_v_ (Si_bulk_/Ag–Si)* is −0.40 eV (=*E_v_ (Si_bulk_)* + Δ*E* = −0.41 eV + 0.01 eV) and that of the valence band of hydrogen-terminated silicon at the silver–silicon contact *E_v_ (Si–H/Ag-Si)* is −0.67 eV (*E_v_* (*Si–H*) + Δ*E* = −0.68 eV + 0.01 eV). The lowest redox level of the Ag^+^/Ag half-cell was 0.45 V vs. SHE (cf. Figure 2a). It would, therefore, be high enough to reach the energetic level of the valence band of the bulk silicon at the silver–silicon contact but not that of the valence band of the hydrogen-terminated silicon. Under this consideration, it would be possible to transfer up to four electronic holes (h_v_^+^) from Ag^+^ reduction to the bulk silicon via the valence band mechanism (Equation (8)).
(8)$$4{\;\mathrm{Ag}}^{+}\;\to \mathrm{Ag}+4{\;h}_{v}{}^{+}\;\mathrm{and}\;{\mathrm{Si}}_{\mathrm{bulk}}+4{\;h}_{v}{}^{+}\;\to {\;\mathrm{Si}}^{4+}$$

If the redox level of the Ag^+^/Ag half-cell exceeds 0.67 V vs. SHE, the energy level of the valence band of the hydrogen-terminated silicon is theoretically reached. This would allow the hole transfer from the Ag^+^ reduction to the valence band of the hydrogen-terminated silicon. The inflection point of the silver deposition kinetics at *E (Ag^+^/Ag)* = 0.65 V vs. SHE marked in Figure 2a deviates only slightly from this theoretical threshold. Based on this consideration, it is assumed that the mean energetic level of the valence band of the hydrogen-terminated silicon at the silver–silicon contact is −0.65 eV. Exceeding this value, the valence transfer from Ag^+^ reduction to bulk silicon and hydrogen-terminated silicon is enabled. Apparently, it is associated with a significant increase in the silver deposition kinetics (*k*_95%,1_ vs. *k*_95%,2_).

Assuming the band bending is due to the differences in the work functions between silver and silicon with Δ*E* = −0.39 eV (=*Φ_Ag_* − *Φ_Si_* = 4.37 eV − 4.76 eV), an energetic level of −0.80 eV for the valence band of the bulk silicon at the silver–silicon contact should result (=*E_v_ (Si_bulk_)* + Δ*E =* −0.41 eV + −0.39 eV) and −1.07 eV (*E_v_ (Si–H)* + Δ*E* = −0.68 eV −0.39 eV) for the valence band of the hydrogen-terminated silicon. With the redox level of the Ag^+^/Ag half-cells between 0.45 V vs. SHE and 0.75 V vs. SHE (Figure 2a), neither valence band is accessible for hole transfer. The process of silver deposition would occur only via the conduction band mechanism. Under the experimental conditions, this mechanism cannot be excluded as the thermally or light-induced excitation of electrons from the valence band into the conduction band of silicon is possible. However, there is no plausible explanation for the inflection point under this consideration. Furthermore, this approach seems to be unlikely, because the higher work function of silicon compared to silver should have resulted in a valence depletion zone on the silver side. Thus, silver would have been the reducing agent and silicon the oxidizing agent. This fundamentally contradicts the experimental findings.

Copper: The argumentation applied to the silver deposition process can also be adapted to copper deposition. If the band bending mechanism is considered based on the differences in the work function of copper (*Φ_Cu_* = 4.57 eV ± 0.53 eV [65,67,68,69]) and silicon, a valence depletion zone would occur on the copper side due to the higher work function of silicon, at an average of 0.19 eV (=*Φ_Si_* − *Φ_Cu_* = 4.76 eV − 4.57 eV). Consequently, copper would be the reducing agent and silicon the oxidizing agent. The redox strength of the Cu^2+^/Cu half-cell would have to exceed a level of at least 0.60 V vs. SHE to match the valence band level of the bulk silicon, which is impossible (*E^0^ (Cu^2+^/Cu)* = 0.34 V vs. SHE [62]).

If the formation of the copper–silicon contact is considered based on a Fermi energy equalization between copper and hydrogen-terminated silicon, the valence and conduction bands of the silicon at the copper–silicon contact are bent by Δ*E* = 0.21 eV (cf. Figure 3b). The energy level of the valence band of bulk silicon is then −0.20 eV (=*E_v_ (Si_bulk_)* + Δ*E* = −0.41 eV + 0.21 eV) and that of the hydrogen-terminated silicon is −0.47 eV (=*E_v_ (Si–H)* + Δ*E* = −0.68 eV + 0.21 eV) at the copper–silicon contact. Under the experimental conditions, the Cu^2+^/Cu half-cells with a redox level between *E (Cu^2+^/Cu)* = 0.20 V vs. SHE (≈*b Cu (diss., t = 0)* ≈ 2.0 × 10^−5^ mol∙kg^−1^) and 0.31 V vs. SHE (cf. Figure 2b) would have been strong enough that valence transfer via the valence band mechanism could occur. The valence transfer from the Cu^2+^/Cu half-cell to the valence band of the hydrogen-terminated silicon would be energetically impossible. Only the Cu^+^/Cu half-cell (*E^0^ (Cu^+^/Cu)* = 0.521 V vs. SHE [62]) reaches the energetic level of *E_v_ (Si–H)* = −0.47 eV.

Two issues are noticeable when examining the kinetics of copper deposition. Firstly, the kinetics at *b Cu (diss., t = 0)* = 2.5 × 10^−4^ mol∙kg^−1^ decrease in a narrow range at a level of *k*_95%_ = 1.1 × 10^−4^ s^−1^ (cf. Figure 2b). This point matches the threshold value of the oxonium ion reaction discussed later in Section 3.6.

The inflection point at *b Cu (diss., t = 0)* = 1.0 × 10^−2^ mol∙kg^−1^ and the associated decrease in the copper deposition rate when it is exceeded are due to the limited silicon dissolution. As mentioned in Equation (5), the dissolution of the oxidized silicon is attributed to either the undissociated HF and/or its anionic dissociation products F^−^ and/or HF_2_^−^ [4,33,36,39,43,53,54,55,56,57,58]. A limitation of HF species should lead to a slower transformation of the oxidized silicon into the soluble product SiF_6_^2−^ and the formation of a more or less thick layer of oxidized silicon, which acts as an insulator [74].

Such a correlation can be mathematically deduced from the activities of the anionic HF dissociation products F^−^, HF_2_^−^ and H_2_F_3_^−^, the amount of silicon dissolution at *t*_95%_ and the diminished metal deposition rates *k*_95%_.

Copper and silver deposition experiments were investigated at three initial levels of total HF concentrations: At *b HF (diss., t = 0)* ≈ 0.06 mol∙kg^−1^, 0.3 mol∙kg^−1^ and 1.5 mol∙kg^−1^. Based on the additions of CuSO_4_-HF or AgF/HF-HF selected and the findings of Messnaoui [75] regarding the molalities of the HF dissociation products in relation to the total HF concentration, the initial molalities of the anionic HF dissociation products F^−^, HF_2_^−^ and H_2_F_3_^−^ were calculated for the individual solutions.

The cumulative initial molalities of these species for the silver batch tests were, depending on the AgF/HF and HF additions, between 7.3 × 10^−3^ and 3.0 × 10^−2^ mol∙kg^−1^ for *b HF (diss., t = 0)* ≈ 0.06 mol∙kg^−1^; between 2.5 × 10^−2^ and 1.4 × 10^−1^ mol∙kg^−1^ at *b HF (diss., t = 0)* ≈ 0.3 mol∙kg^−1^; and between 1.6 × 10^−1^ and 2.8 × 10^−1^ mol∙kg^−1^ at *b HF (diss., t = 0)* ≈ 1.5 mol∙kg^−1^. The Cu(II)/HF solutions were based on independent additions of Cu^2+^ and HF. Therefore, the cumulative initial molalities of F^−^, HF_2_^−^ and H_2_F_3_^−^ were identical: 7.4 × 10^−3^ mol∙kg^−1^ for the batches with *b HF (diss., t = 0)* ≈ 0.06 mol∙kg^−1^, 2.6 × 10^−2^ mol∙kg^−1^ with *b HF (diss., t = 0)* ≈ 0.3 mol∙kg^−1^ and 1.6 × 10^−1^ mol∙kg^−1^ with *b HF (diss., t = 0)* ≈ 1.5 mol∙kg^−1^.

Based on the Bromley equations [64], the mean activity coefficients of the anionic HF species *γ (F^−^, HF_2_^−^ and H_2_F_3_^−^)* were calculated. The *γ (F^−^, HF_2_^−^ and H_2_F_3_^−^)* value for AgF/HF solutions was = 0.91 for the solution with the lowest ionic strength of *I* = 7.3 × 10^−3^ mol∙kg^−1^, and *γ (F^−^, HF_2_^−^ and H_2_F_3_^−^)* was 0.66 for the solution with the highest ionic strength of *I* = 2.8 × 10^−1^ mol∙kg^−1^. In the Cu^2+^/HF solutions, *γ (F^−^, HF_2_^−^ and H_2_F_3_^−^)* varied between 0.69 in the lowest concentrated solution (*I* = 7.5 × 10^−3^ mol∙kg^−1^) and 0.05 in the highest concentrated solution (*I* = 1.8 × 10^0^ mol∙kg^−1^). The activities of the anionic HF species *a (F^−^, HF_2_^−^ and H_2_F_3_^−^) (t = 0)* were calculated from the product of the initial molalities and the mean activity coefficients calculated. The results for the individual solutions are plotted against the initial dissolved molalities of Ag^+^ and Cu^2+^ in the left half of Figure 4a,b. The molalities of the dissolved silicon at *t*_95%_
*b Si (diss., t_95%_)* are shown in the same diagrams. The quotient of both parameters in relation to the initial molalities of Ag^+^ and Cu^2+^ is given in the right half of Figure 4a,b. It is recognizable that more silicon was oxidized and converted into the dissolved form as a result of the increasing initial molality of Ag^+^ and Cu^2+^. If the ratio of *a (F^−^, HF_2_^−^ and H_2_F_3_^−^) (t = 0)* and *b Si (diss., t_95%_)* was <6:1 mol∙kg^−1^:mol∙kg^−1^, there was a significant decrease of the metal deposition rates *k*_95%_ compared to the experiments with *a (F^−^, HF_2_^−^ and H_2_F_3_^−^) (t = 0)*:*b Si (diss., t_95%_)* ratio of ≥6:1 mol∙kg^−1^:mol∙kg^−1^. This criterion found reflects the stoichiometric ratio of F and Si of the formal end product of the oxidized silicon dissolution SiF_6_^2−^. This leads to the conclusion that the species F^−^, HF_2_^−^ and H_2_F_3_^−^ must be responsible for the rapid dissolution of the oxidized silicon.

The data points with limited F^−^, HF_2_^−^ and H_2_F_3_^−^ activity are marked in gray in Figure 4a,b and in Figure 2a,b for visualization. The comparison of the metal deposition rates *k*_95%_ shown in Figure 2a,b and the *a (F^−^, HF_2_^−^ and H_2_F_3_^−^) (t = 0)*:*b Si (diss., t_95%_)* ratios in Figure 4a,b indicates that the smaller the *a (F^−^, HF_2_^−^ and H_2_F_3_^−^) (t = 0)*:*b Si (diss., t_95%_)* ratio was, the more the kinetics of metal deposition diminished. If the metal deposition rates *k*_95%_ were set as a reference at *a (F^−^, HF_2_^−^ and H_2_F_3_^−^) (t = 0)*: *b Si (diss., t_95%_)* ratio of ≥ 6:1 mol∙kg^−1^:mol∙kg^−1^, the kinetics of metal deposition at an *a (F^−^, HF_2_^−^ and H_2_F_3_^−^) (t = 0)*:*b Si (diss., t_95%_)* ratio of 3:1 mol∙kg^−1^:mol∙kg^−1^ at nominally the same initial Ag^+^ or Cu^2+^ molality was reduced by about half. At an *a (F^−^, HF_2_^−^ and H_2_F_3_^−^) (t = 0)*:*b Si (diss., t_95%_)* ratio of 1:1 mol∙kg^−1^:mol∙kg^−1^, the kinetics of metal deposition was only about 1/10 of that without a calculated limitation of the activities of the anionic fluoride species.

The kinetic inhibition of metal deposition due to a restricted access of HF species to the silicon surface caused by the deposited metal film itself, postulated by Geyer et al. [33], does not seem to have been the decisive criterion for the experiments of this study. Otherwise, the metal deposition rates *k*_95%_ of all metal depositions should have decreased in unison with the increasing initial metal ion molality (cf. Figure 2). Furthermore, it was also observed that the metal layers in all experiments had partially detached from the wafers during the deposition process due to the low adhesion forces between metal and silicon [6,25,26,27] and the continuous stirring of the solutions. As a result, the silicon surface was repeatedly in contact with the HF solution.

Gold: Two inflection points of kinetics have been identified in the gold deposition experiments (cf. Figure 2c). In contrast to the findings for copper, the inflection points were not due to a limitation of the anionic HF dissociation products. The *a (F^−^ + HF_2_^−^ + H_2_F_3_^−^) (t = 0)*:*b Si (diss., t_95%_)* ratios were above the critical limit of 6:1 mol∙kg^−1^:mol∙kg^−1^, with the exception of the experiment with the highest initial Au(III) molality of *b Au (diss., t = 0)* = 9.3 × 10^−2^ mol∙kg^−1^. It is obvious that the kinetics of gold deposition are a mirror image of those of the silver deposition. The functional increase in the gold deposition rate is potential between *b Au (diss., t = 0)* = 3.0 × 10^−7^ mol∙kg^−1^ (*E (AuCl_4_^−^/Au)* = 0.87 V vs. SHE) and 3.1 × 10^−5^ mol∙kg^−1^ (*E (AuCl_4_^−^/Au)* = 0.91 V vs. SHE), analogously to the kinetics of silver deposition above the level of *b Ag (diss., t = 0)* = 6.0 × 10^−3^ mol∙kg^−1^ or *E (Ag^+^/Ag)* = 0.65 V vs. SHE (cf. Figure 2a,c). The redox levels of the Me^z+^/Me half-cells in both cases are strong enough that holes can be transferred from the metal side to the valence band of bulk silicon and hydrogen-terminated silicon (cf. Figure 2a,c and Figure 3a,c). Exceeding the first inflection point, the gold deposition kinetics drop to a minimum at *b Au (diss., t = 0)* = 1.5 × 10^−5^ mol∙kg^−1^. Above this point, the gold deposition rate increased up to a level which was determined for silver below the threshold of *b Ag (diss., t = 0)* = 6.0 × 10^−3^ mol∙kg^−1^ or *E (Ag^+^/Ag)* = 0.65 V vs. SHE.

The behavior of gold deposition rates indicates a partial inhibition of the valence transfer processes when the inflection point is exceeded. The amount of band bending at the gold–silicon contact in the gold deposition experiments is the decisive parameter for the kinetics (cf. Figure 3c). According to the theoretical values of the work functions of gold (*Φ_Au_* = 4.96 eV ± 0.51 eV [65,67,68,69] and hydrogen-terminated silicon in vacuum, the valence and conduction bands would have to be bent by an average of Δ*E* = +0.60 eV after Fermi energy equalization at the gold–silicon contact. This means a theoretical energy level of +0.19 eV at the gold–silicon contact (=*E_v_ (Si_bulk_)* + Δ*E* = −0.41 eV + 0.60 eV) for the valence band of bulk silicon. This provides a potential barrier for the valence transfer between gold and silicon along the Fermi energy level to the valence band of the bulk silicon. It is known as the Schottky barrier *E_sb_*. Thus, the valence transfer between AuCl_4_^−^ and bulk silicon is inhibited, while the valence transfer via the valence band mechanism with the hydrogen-terminated silicon is accessible, since with the band bending, the energy level of the valence band of the hydrogen-terminated silicon at the gold–silicon contact reaches a theoretical energy level of −0.08 eV (=*E_v_ (Si–H)* + Δ*E* = −0.68 eV + 0.60 eV). According to the findings for the gold deposition rates (Figure 2c), the partial inhibition of the valence transfer claimed occurs above an initial AuCl_4_^−^ molality of *b Au (diss., t = 0)* = 3.1 × 10^−5^ mol∙kg^−1^ and is assumed to be fully effective above an initial AuCl_4_^−^ molality of *b Au (diss., t = 0) =* 1.5 × 10^−4^ mol∙kg^−1^.

The proposed dependence of the potential barrier on the initial AuCl_4_^−^ molality means that the amount of band bending at the metal–silicon contact must correlate with *b Au (diss., t = 0)*. Therefore, there is a requirement that the difference in work functions between gold and silicon is also positively correlated with the AuCl_4_^−^ molality. This is possible because the effective work function is not only a material property. It is influenced by the chemical potential of the supernatant solution, and thus, in these experiments and others, by the AuCl_4_^−^ molality. The effect of the adsorption of ions on the metal surface becomes relevant if a significant dipole moment is generated at the boundary layer of the metal surface, which impedes the valence transfer from the solution through the metal to the silicon resulting in an enhanced effective work function. Gossenberger et al. [60] have shown, for example, that a fluoride adsorption of the platinum surface causes a strong dipole moment on the metal surface, which increases the effective work function of platinum by about 2 eV when the metal is almost completely covered with fluoride. Based on the kinetics of gold deposition, a similar effect by the adsorption of AuCl_4_^−^ and/or its reduction intermediates on the gold surface is assumed. Based on the hypothesis that valence transfer to the valence band of the hydrogen-terminated silicon was not affected by a potential barrier, the effective work function of the gold may have been increased by a maximum of Δ*E* = 0.68 eV (*E_v_ (Si–H)* + Δ*E* = −0.68 eV + 0.68 eV = 0 eV = *E_F_*) over the work function of hydrogen-terminated silicon (cf. Figure 3c).

Platinum: The kinetics of platinum deposition (Figure 2d) are quite similar to those of gold deposition up to the inflection points. However, the platinum deposition rates *k*_95%,1_ are lower than the analogous gold deposition rates relative to the initial metal ion molalities. This is due to the fact that the PtCl_6_^2−^/Pt half-cell is weaker than the AuCl_4_^−^/Au half-cell (cf. Figure 2c,d). The valence bands of the bulk and hydrogen-terminated silicon are hypothetically accessible for valence transfers until the inflection points. In contrast to gold deposition, the platinum deposition rates *k*_95%,2_ decrease logarithmically to the initial PtCl_6_^2−^ molality or the redox level of the PtCl_6_^2−^/Pt half-cell after the second inflection point at *b Pt (diss., t = 0)* = 6.0 × 10^−5^ mol∙kg^−1^ has been exceeded. This would be plausible if Schottky barriers were constituted at the platinum–silicon interface above this level limiting the hole transfer to the valence band of bulk silicon and that of hydrogen-terminated silicon at higher PtCl_6_^2−^ molalities. The energy scheme at the platinum–silicon interface shown in Figure 3d indicates this assumption. The average work function of platinum of *Φ_Pt_* = 5.53 ± 0.40 eV in vacuum [65,67,68,69] is higher by an average of Δ*E* = 1.17 eV compared to the average work function of the hydrogen-terminated silicon. The band bending results in an energy level of the valence band of the bulk silicon at the platinum–silicon contact of +0.76 eV (=*E_v_ (Si_bulk_)* + Δ*E* = −0.41 eV + 1.17 eV). The energy of the valence band of the hydrogen-terminated silicon has a value of +0.49 eV (=*E_v_ (Si–H)* + Δ*E* = −0.68 eV + 1.17 eV). Kuznetsov et al. [47] have experimentally determined a barrier height of up to 0.30 eV at a platinum–silicon contact in HF solution. The effect of the Schottky barrier on the kinetics of metal deposition for platinum is greatest compared to all other metals because platinum has the highest work function of all metals.

### 3.4. Stoichiometry of Metal Cation Reduction and Silicon Oxidation at Solitary Initial Metal Ion and HF Concentration Levels

In addition to the consideration of the kinetics, the stoichiometric ratio of metal deposition Δ*b Me* and silicon dissolution Δ*b Si* can be determined from the experimental data. The ratio of the decrease of dissolved metal ion molality and that of the increase of silicon molality from *t* = 0 to *t*_95%_ (Δ*b Me:* Δ*b Si (diss., t_95%_)*) was calculated for the four examples shown in Figure 1 according to Equation (9). This calculation approach provided the time-weighted average of the stoichiometric ratio.
(9)$$\frac{\Delta b\;Me\;}{\Delta b\;Si}\left(diss.,\;t_{95\%}\right)=\frac{b\;Me\;\left(diss.,t\;=\;0\right)-b\;Me\;\left(diss.,t_{95\%}\right)}{b\;Si\;\left(diss.,t_{95\%}\right)-b\;Si\;\left(diss.,t\;=\;0\right)}$$

Different Δ*b Me*: Δ*b Si (diss., t_95%_)* ratios were obtained at roughly the same initial metal ion molalities of about 1 × 10^−2^ mol∙kg^−1^. The ratios were Δ*b Ag*:Δ*b Si (diss., t_95%_)* ≈ 3.7:1 mol∙kg^−1^:mol∙kg^−1^, Δ*b Cu*:Δ*b Si (diss., t_95%_)* ≈ 1.5:1 mol∙kg^−1^:mol∙kg^−1^, Δ*b Au*:Δ*b Si (diss., t_95%_)* ≈ 1.0:1 mol∙kg^−1^:mol∙kg^−1^ and Δ*b Pt*:Δ*b Si (diss., t_95%_)* ≈ 1.0:1 mol∙kg^−1^:mol∙kg^−1^. The valence balance derived from the reaction schemes formulated in Equations (1)–(3) implies that three valences would have been formally transferred in copper and gold deposition, between three and four valences in silver deposition and four valences would have been transferred in platinum deposition during metal ion reduction to the metal state for the oxidation of one silicon atom. Thus, theoretically, between zero and one valence would be missing to complete the balance regarding silicon oxidation up to the oxidation state of +IV in the final product SiF_6_^2−^. Under this convention, at least one additional oxidant would have to be involved in the silicon oxidation to complete the missing part of the valence transfer. According to the current state of knowledge in the literature, this is attributed to the reduction of hydrogen ions to molecular hydrogen [16,38,39,40,43,49,50,51,52] according to the reaction equations formulated in Equations (2) and (4).

The changes in the molalities of the dissolved metal and silicon species of the experiments presented in Figure 1 were calculated with a different approach based on the results of the consecutive samples analogous to Equation (9). The time-dependent stoichiometric ratios calculated are shown in Figure 5. It is noticeable that the stoichiometric ratios are not constant in the period of *t* = 0 to *t*_95%_ and beyond and not similar for the different metal deposits. The ranges of the stoichiometric Δ*b Me*:Δ*b Si* ratios calculated indicate that between two (lower limit for copper deposition) and 4.8 valences (upper limit for platinum deposition) per dissolved silicon atom must have been formally exchanged. As can be deduced by the example of silver deposition above *t*_95%_, the stoichiometric ratio of Δ*b Ag*:Δ*b Si* ≈ 0:1 mol∙kg^−1^:mol∙kg^−1^ also reveals that no valence transfer by metal reduction can cause the silicon oxidation and consecutive silicon dissolution. In conclusion, there must be at least one additional oxidizing agent besides metal cations and at least one additional reducing agent besides silicon in order to achieve a complete valence balance.

Including the data of the other experiments, it can be stated that the metal-specific sequences of the stoichiometric Δ*b Me*:Δ*b Si* ratios are, in most cases, similar to the examples shown in Figure 5.

Silver: The initial stoichiometric Δ*b Ag*:Δ*b Si* ratio (between *t* = 0 and *t* = 300 s) in the silver deposition experiments was mostly about 75% of the maximum Δ*b Ag*:Δ*b Si* ratio. Subsequently, the stoichiometric ratios increased up to the respective maximum. The higher the initial Ag^+^ molality, the later the maximum of the respective Δ*b Ag*:Δ*b Si* ratio was reached. It was achieved at *t* = 600 s in the example of Figure 5a. Subsequently, the Δ*b Ag*:Δ*b Si* ratios decreased mostly below the initial level. The Δ*b Ag*:Δ*b Si* ratio in a total of nine experiments dropped sharply after the molality of *b Ag (diss., t = i)* had decreased to a level below 3.3 × 10^−4^ mol∙kg^−1^ (cf. Figure 1a and Figure 5a, between *t* = 2400 s and *t* = 3600 s).

Copper: Copper deposition usually started with a Δ*b Cu*:Δ*b Si* ratio at about 50% of the maximum Δ*b Cu*:Δ*b Si* ratio. Contrary to the silver deposition, the higher the initial Cu^2+^ molality, the earlier the maximum of Δ*b Cu*:Δ*b Si* occurred. It was achieved at *t* = 1800 s in the example of Figure 5b. Afterwards, the Δ*b Cu*:Δ*b Si* ratios decreased again to about the level of the initial stoichiometric ratio. Analogously to the silver experiments, there seems to be a threshold value at *b Cu (diss., t = i)* ≈ 2.5 × 10^−4^ mol∙kg^−1^, below which the Δ*b Cu*:Δ*b Si* ratio is significantly reduced. This value corresponds to the finding of the kinetics at which the copper deposition rate *k*_95%_ has reached a local minimum (cf. Figure 2b).

Gold: Gold depositions started with one exception, as shown in Figure 5c, with the highest Δ*b Au*:Δ*b Si* ratio at *t* = 300 s. After that, the Δ*b Au*:Δ*b Si* ratios potentially decreased in relation to the time *t = i*. The experiment with the highest initial AuCl_4_^−^ concentration of 9.3 × 10^−3^ mol∙kg^−1^ shows a reversed temporal sequence of the Δ*b Au*:Δ*b Si* ratios. Analogously to silver and copper deposition, when the Δ*b Au*:Δ*b Si* ratio falls below a threshold of *b Au (diss., t = i)* ≈ 1.5 × 10^−4^ mol∙kg^−1^, there also appears to be a significant decrease in the Δ*b Au*:Δ*b Si* ratio compared to the previous Δ*b Au*:Δ*b Si* ratios (cf. *t* = 0–*t* = 2400 s vs. *t* = 2400 s–*t* = 3600 s in Figure 1c and Figure 5c). This value corresponds to the second kinetic inflection point of gold deposition marked in Figure 2c.

Platinum: The platinum deposition experiment shown in Figure 5d is the exception in this series. In this example, the Δ*b Pt*:Δ*b Si* ratio started at about 60% of the maximum ratio, decreased at *t* = 2400 s to the minimum to about 50% of the maximum Δ*b Pt*:Δ*b Si* ratio and then increased potentially to the maximum Δ*b Pt*:Δ*b Si* ratio of 6:5 mol∙kg^−1^:mol∙kg^−1^ at *t*_95%_. The sequence of the Δ*b Pt*:Δ*b Si* ratios looks mirror-inverted to the abscissa *t* for the platinum deposits with initial PtCl_6_^2−^ molalities of less than 1.0 × 10^−4^ mol∙kg^−1^. The initial Δ*b Pt*:Δ*b Si* ratio started at 40% of the respective maximum ratio, reached its maxima at *t* = 2400 s and then fell with potential progression to a level below the initial Δ*b Pt*:Δ*b Si* ratio. The stoichiometric Δ*b Pt*:Δ*b Si* ratio decreased significantly in two experiments when the PtCl_6_^2−^ molality fell below a threshold of *b Pt (diss., t = i)* ≈ 6.0 × 10^−4^ mol∙kg^−1^. Similar to gold deposition, this value corresponds to the second kinetic inflection point marked in Figure 2d.

### 3.5. Stoichiometry of Metal Cation Reduction and Silicon Oxidation at Different Initial Metal Ion and HF Concentration Levels

Figure 6 summarizes the Δ*b Me*:Δ*b Si* ratios of all metal deposits in relation to the initial molality of the metal ion *b Me (diss., t = 0)* dissolved. The diagrams show the *t*_95%_-weighted stoichiometric Δ*b Me*:Δ*b Si* ratios over the period from *t* = 0 to *t*_95%_ (Δ*b Me*:Δ*b Si (diss., t_95%_)*), as shown in Figure 5 as points and the variations of the stoichiometric ratios within this period as bars. The results of Figure 5 have been colored red in Figure 6.

It can be recognized that a higher initial metal ion molality is linked with a larger stoichiometric Δ*b Me*:Δ*b Si (diss., t_95%_)* ratio, but the Δ*b Me*:Δ*b Si (diss., t_95%_)* ratios are limited at the lower and upper end of the range of initial metal ion molarities. The Δ*b Me*:Δ*b Si* ratios related to the *t*_95%_-weighted findings are just over 0:1 mol∙kg^−1^:mol∙kg^−1^ at the initial metal ion molalities of less than 1.0 × 10^−5^ mol∙kg^−1^. The upper limits are metal-specific. The *t*_95%_-weighted Δ*b Ag*:Δ*b Si* ratio for the silver deposition is a maximum of 4.3:1 mol∙kg^−1^:mol∙kg^−1^. The analog maximum ratio for the copper deposition is 1.9:1 mol∙kg^−1^:mol∙kg^−1^. The maximum Δ*b Au*:Δ*b Si* ratio for gold deposition is about 1.1:1 mol∙kg^−1^:mol∙kg^−1^, and the maximum ratio for platinum deposition is about 1.0:1 mol∙kg^−1^:mol∙kg^−1^, as shown in Figure 5d.

As discussed in Figure 5, the maxima and minima of the Δ*b Me*:Δ*b Si* ratios between *t* = 0 and *t*_95%_ differ significantly from the *t*_95%_-weighted ratios. Sigmoidal functional relationships were calculated for the maxima and minima in relation to the initial molalities of the metal ion and are shown in Figure 6. The functions *f*_1_ reflect the course of the maxima. The latter was calculated based on Equation (10). The parameter *(*Δ*b Me*:Δ*b Si)_max_* included represents the absolute maximum of the stoichiometric ratios of Δ*b Me*:Δ*b Si*. *(*Δ*b Ag*:Δ*b Si)_max_* is ≈ 4:1 mol∙kg^−1^:mol∙kg^−1^ up to *b Ag (diss., t = 0)* = 5.5 × 10^−3^ mol∙kg^−1^ for the silver deposition. Above this threshold, *(*Δ*b Ag*:Δ*b Si)_max_* is 5:1 mol∙kg^−1^:mol∙kg^−1^. There are also two maxima for the copper deposition. The first maximum of *(*Δ*b Cu*:Δ*b Si)_max_* is 2:1 mol∙kg^−1^:mol∙kg^−1^ up to a threshold of *b Cu (diss., t = 0)* = 8.2 × 10^−3^ mol∙kg^−1^. The second maximum of Δ*b Cu*:Δ*b Si)_max_* = 3:1 mol∙kg^−1^:mol∙kg^−1^ is located above. Regarding the platinum deposition, *f*_1_ converges to a maximum of 1:1 mol∙kg^−1^:mol∙kg^−1^ up to *b Pt (diss., t = 0)* = 5.1 × 10^−5^ mol∙kg^−1^ and above that to a maximum of 6:5 mol∙kg^−1^:mol∙kg^−1^. There is one maximum *(*Δ*b Au*:Δ*b Si)_max_* = 4:3 mol∙kg^−1^:mol∙kg^−1^ for the gold deposition. With an approximation of *b Me (diss., t = 0)* to the value of 0 mol∙kg^−1^, *f*_1_ converges towards a Δ*b Me*:Δ*b Si* ratio of 0:1 mol∙kg^−1^:mol∙kg^−1^. The functional increase between the lower and upper limits is described by the empirical term *x∙b Me (diss., t = 0)^0.01^*. The validity ranges for *f*_1_ regarding the initial metal ion molality *b Me (diss., t = 0)*, the absolute maxima *(*Δ*b Me*:Δ*b Si)_max_* and the empirical slope parameter *x*, can be found in Table 1.
(10)$$\begin{array}{cc} f_{1}\mathrm{or}\;f_{2}=\frac{\Delta b\;Me}{\Delta b\;Si} & \left(diss.,t_{95\%}\right)\\ & =\frac{b\;Me\;\left(diss.,\;t\;=\;0\right)\;}{\frac{b\;Me\;\left(diss.,\;t\;=\;0\right)\;}{{\left(\frac{\Delta b\;Me}{\Delta b\;Si}\right)}_{max}\;}+x\;\cdot b\;Me\;{\left(diss.,\;t\;=\;0\right)}^{0.01}} \end{array}$$

Analogously to the maximum function *f*_1_, the functions of the minimas *f*_2_ were calculated according to Equation (10). The levels of the Δ*b Me*:Δ*b Si* minima ratios increase similarly to the maxima with increasing initial metal ion molalities *b Me (diss., t = 0)*. The functions *f*_2_ converge to the maxima of the functions *f*_1_ for copper, gold and platinum deposition (the first maximum of 2:1 mol∙kg^−1^:mol∙kg^−1^ for copper deposition, and the second maximum of 6:5 mol∙kg^−1^:mol∙kg^−1^ for platinum deposition). The functions *f*_1_ and *f*_2_ differ in the slope parameter *x*. By deviation, the minimum function *f*_2_ of the silver deposition converges to a lower maximum ratio compared to the *f*_1_ function with *(*Δ*b Ag*:Δ*b Si)_max_* = 10:3 mol∙kg^−1^:mol∙kg^−1^. The parameterization of *f*_2_ is also shown in Table 1.

The exceeding of the level of *b Me (diss., t = 0)* between 6.5 × 10^−4^ mol∙kg^−1^ (Pt) and 2.4 × 10^−3^ mol∙kg^−1^ (Au) resulted in a significant deviation from the minimum functions *f*_2_ in the copper, gold and platinum deposition series (but not in the silver deposition). Instead of a convergence of the functions towards the Δ*b Me*:Δ*b Si* maxima ratios of the *f*_1_ functions, an inverse sigmoidal course of the Δ*b Me*:Δ*b Si* ratios corresponding to the function *f_3_* according to Equation (11) occurred (parameterization in Table 1). This means that the minima of the Δ*b Me*:Δ*b Si* ratios decreased with higher initial metal ion molalities *b Me (diss., t = 0)* towards an absolute minimum ratio *(*Δ*b Me*:Δ*b Si)_min_*. These minimum stoichiometric ratios are 1:2 mol∙kg^−1^:mol∙kg^−1^ for copper and platinum deposition and 1:3 mol∙kg^−1^:mol∙kg^−1^ for gold deposition.
(11)$$\begin{array}{c} f_{3}=\frac{\Delta b\;Me}{\Delta b\;Si}\left(diss.,t_{95\%}\right)\\ ={\left(\frac{\Delta b\;Me}{\Delta b\;Si}\right)}_{max,f_{2}}\\ -\frac{b\;Me\;\left(diss.,\;t\;=\;0\right)\;}{\frac{b\;Me\;\left(diss.,\;t\;=\;0\right)\;}{{\left(\frac{\Delta b\;Me}{\Delta b\;Si}\right)}_{max,f_{2}}-{\left(\frac{\Delta b\;Me}{\Delta b\;Si}\right)}_{min}}+x\cdot b\;Me\;{\left(diss.,\;t\;=\;0\right)}^{0.01}} \end{array}$$

Based on these results, certain analogous reaction mechanisms can be derived for all metal depositions. The various findings regarding the stoichiometric Δ*b Me*:Δ*b Si* ratios are indications of the involvement of different oxidizing and reducing agents. In connection with the conclusions drawn from the kinetic data about the involvement of bulk and hydrogen-terminated silicon, four sections of the process of metal deposition on silicon can be interpreted, denoted as *I, II, III* and *IV* in Figure 6.

### 3.6. Interpretation of the Stoichiometric Findings

#### 3.6.1. The Involvement of the Oxonium Ion (*Section I*)

The stoichiometric Δ*b Me*:Δ*b Si* ratios are close to 0:1 mol∙kg^−1^:mol∙kg^−1^ for the initial metal ion molalities between 1.0 × 10^−7^ and ≈ 1.0 × 10^−5^ mol∙kg^−1^. Regarding the valence balance, this finding means that silicon oxidation up to the oxidation state of +IV cannot have been achieved by reducing the metal cations alone. It is assumed that the oxonium ions H_3_O^+^ resulting from the HF dissociation in water must have been involved in the oxidation process of the silicon up to a certain *b Me (diss., t = 0)* threshold. A comparison of the standard potential of the 2H_3_O^+^/H_2_ half-cell of 0 V vs. SHE [62] with the energy levels of the valence bands of the bulk and hydrogen-terminated silicon (*E_v_ (Si_bulk_)* = −0.41 eV and *E_v_ (Si–H)* = −0.68 eV), outlined in Figure 3, indicates that a valence exchange initiated via the valence band mechanism could never have occurred without the band bending mechanism at the metal–silicon contact. Taking into account the band bending mechanism and the standard potential of the 2H_3_O^+^/H_2_ half-cell of 0 V vs. SHE [62], the oxonium ion reduction would only be possible in the presence of the gold-silicon or platinum–silicon contact (cf. Figure 3c,d). The energy levels of the silicon valence bands at the silver–silicon and copper–silicon contacts are between 0.20 eV (*E_v_ (Si_bulk_/Cu-Si)*) and 0.67 eV (*E_v_ (Si-H/Ag-Si)*) above the energy level of the standard 2H_3_O^+^/H_2_ half-cell (cf. Figure 3a,b). From this point of view, the oxonium ion reduction has to be excluded for the silver and copper deposition experiments. However, as a result of the similar Δ*b Me*:Δ*b Si* ratios in all metal deposits tested, the oxonium ion-based process is assumed for all kinds of metal depositions. The supposed paradox can be explained by the underpotential or overvoltage kinetic effect of molecular hydrogen formation on metal surfaces.

If a critical current density *J* of metal cations on the metal surface is undershot, a positive equilibrium potential is present (underpotential effect), and conversely, a negative equilibrium potential (overvoltage) of the 2H_3_O^+^/H_2_ half-cell on the metal surface is established. The amount of deviation from the standard potential of the 2H_3_O^+^/H_2_ half-cell depends on the metal and the current density. The correlation was determined experimentally by Hunt et al. [76] for Ag, Cu, Au and Pt electrodes. The current density *J* (A∙cm^−1^) for the experiments of this study was roughly calculated based on Equation (12). The numerator is the product of the initial metal ion molality *b Me (diss, t = 0)*, the mass of the solution *m_sol_* (=0.06 kg), the factor 0.95 (because of the meaning of *t*_95%_), the charge number *z* for Ag(I), Cu(II), Au(III) or Pt(IV) and the Faraday constant *F* (96485 A∙s∙mol^−1^); the denominator is the product of the metal surface area *A_Me_*, which was roughly equated with the silicon wafer surface area *A_Si_* (=46 cm²) and *t*_95%_.
(12)$$J=\frac{b\;Me\;\left(diss.,\;t\;=\;0\right)\cdot m_{sol}\cdot 0.95\cdot z\cdot F}{A_{Me}\cdot t_{95\%}}$$

The equilibrium potentials of the 2H_3_O^+^/H_2_ half-cells were derived from the findings of Hunt et al. [76] in comparison with the current densities *J* calculated. The averaged values of the equilibrium potentials given for bright metal electrodes and metallic colloids in the solutions in this publication served as the basis for the calculation. This approach is in accordance with the observations of the experiments in this study, whereas the metals were deposited on the surface and particles of the metal were present in the solutions due to partial metal detachment from the silicon surface. The equilibrium potentials of the 2H_3_O^+^/H_2_ half-cells determined are shown in Figure 7 in relation to the initial metal ion molalities *b Me (diss., t = 0)*. The equilibrium potentials of the 2H_3_O^+^/H_2_ half-cells can also be found in the energy diagrams in Figure 3. In turn, the findings of Figure 3 regarding the energetic level of the valence bands of the bulk and hydrogen-terminated silicon and the Fermi energy *E_F_* at the metal–silicon contact were transferred to Figure 7. In addition, the initial metal ion molalities *b Me (diss., t = 0)* were drawn in Figure 7, and have the intersection with *f*_1_ at Δ*b Me*:Δ*b Si* = 3:z mol∙kg^−1^:mol∙kg^−1^ and with *f*_2_ at Δ*b Me*:Δ*b Si* = 1:z mol∙kg^−1^:mol∙kg^−1^, according to Figure 6. It is assumed that oxonium ion reduction occurs below and up to these threshold values. The range of the oxonium ion effect has been marked with *I* in Figure 6 and Figure 7.

According to the *f*_1_ and *f*_2_ functions shown in Figure 6a, the threshold for silver deposition is *b Ag (diss., t = 0)* = 3.3 × 10^−4^ mol∙kg^−1^.This is the value identified at which the Δ*b Ag*:Δ*b Si* ratios decreased significantly when it was undershot during silver deposition experiments (cf. Figure 1a and Figure 5a). Figure 7a shows that the equilibrium potentials of the 2H_3_O^+^/H_2_ half-cell below this threshold are above the energy level of the valence band of bulk silicon at the silver–silicon contact but below the energy level of the valence band of hydrogen-terminated silicon (hollow symbols). This means that the oxonium ion reaction with the bulk silicon can occur due to the underpotential effect of the 2H_3_O^+^/H_2_ half-cell on the silver surface. The reaction with the hydrogen-terminated silicon is not possible. If the threshold value mentioned is exceeded, the 2H_3_O^+^/H_2_ half-cell is too weak to oxidize the bulk silicon via the valence band mechanism (full symbols). The stoichiometric Δ*b Ag*:Δ*b Si* ratio at the threshold value is a maximum of 3:1 mol∙kg^−1^:mol∙kg^−1^ (*f*_1_) and a minimum of 1:1 mol∙kg^−1^:mol∙kg^−1^ (*f*_2_). According to the valence balance, a maximum of three valences were exchanged by the reduction of 3 Ag^+^ to 3 Ag with one atom of the bulk silicon and one valence was exchanged by the reduction of one oxonium ion to elemental H with the oxidation state ±0. At the minimum, three valences were transferred to one silicon atom of the bulk silicon by oxonium ion reduction and one valence by the reduction of Ag^+^.

The findings for copper deposition are similar (cf. Figure 7b). The threshold value of *b Cu (diss., t = 0)* derived from Figure 6b is 2.5 × 10^−4^ mol∙kg^−1^. This threshold value was also identified in the copper deposition experiments as the value below which the Δ*b Cu*:Δ*b Si* ratio has been significantly diminished. Furthermore, it is the point at which the copper deposition rate has reached an intermediate minimum (cf. Figure 2b). The maximum Δ*b Cu*:Δ*b Si* ratio calculated at *b Cu (diss., t = 0)* = 2.5 × 10^−4^ mol∙kg^−1^ was 3:2 mol∙kg^−1^:mol∙kg^−1^ (*f*_1_) and the minimum Δ*b Cu*:Δ*b Si* ratio was 1:2 mol∙kg^−1^:mol∙kg^−1^ (*f*_2_). Similarly to silver deposition, this means one up to three valences must have been transferred from the oxonium ion reduction forming H (±0), and three up to one valences must have been transferred from Cu^2+^ reduction to Cu state to one silicon atom of the bulk silicon. According to the higher energetic level of the valence band of the hydrogen-terminated silicon, no reaction with the hydrogen terminated silicon is possible.

The experimental threshold value of *b Au (diss., t = 0)* observed and calculated for gold deposition is 1.5 × 10^−4^ mol∙kg^−1^ (Figure 7c). Similarly to copper deposition, this point corresponds to the intermediate minimum of the gold deposition rate (Figure 2c). The Δ*b Au*:Δ*b Si* intersection point with the *f*_1_ function is 1:1 mol∙kg^−1^:mol∙kg^−1^ and that of the *f*_2_ function is 1:3 mol∙kg^−1^:mol∙kg^−1^ (cf. Figure 6c). The valence balance is similar to the silver and copper deposition findings. In contrast to the silver and copper deposition, the reaction of the oxonium ions with the bulk and hydrogen-terminated silicon is possible, due to the relatively strong underpotential effect of the 2H_3_O^+^/H_2_ half-cell on the gold surface and the strong silicon valence band bending at the gold–silicon contact (cf. Figure 3c). Above the threshold value, the equilibrium potential of the 2H_3_O^+^/H_2_ half-cell decreases to <0 V vs. SHE and the oxonium ion-based reaction is no longer possible. Firstly, it is assumed that the hydrogen termination of the silicon is process limiting. It acts hypothetically like a standard 2H_3_O^+^/H_2_ half-cell, and thus limits the valence transfer to the silicon. Secondly, the Fermi energy level of the gold–silicon contact cannot be reached by the 2H_3_O^+^/H_2_ half-cell with a redox level of less than 0 V vs. SHE. This is the essential condition for the transfer of the electronic holes to the silicon via the valence band mechanism.

Basically, the same applies to platinum deposition (Figure 7d). The value observed experimentally and the threshold value for the oxonium ion reaction with silicon determined functionally is *b Pt (diss., t = 0)* = 6.0 × 10^−5^ mol∙kg^−1^ (cf. Figure 6d). This point corresponds to the second inflection point determined in the platinum deposition rate (cf. Figure 2d). At this point, the maximum Δ*b Pt*:Δ*b Si* ratio is 3:4 mol∙kg^−1^:mol∙kg^−1^ and the minimum Δ*b Pt*:Δ*b Si* ratio is 1:4 mol∙kg^−1^:mol∙kg^−1^. The valence balance for the platinum–silicon system at this threshold is identical to the silver, copper and gold deposition findings at their respective thresholds. Based on this knowledge, the reaction process within *section I* can be summarized according to the scheme in Equation (13).
(13)$$\begin{array}{cc} \left(>0..\frac{3}{z}\right)\;{\mathrm{Me}}^{z+}+ & \left(<4..1\right)H_{3}O^{+}+\mathrm{Si}\\ & ⟶\left(>0..\frac{3}{z}\right)\mathrm{Me}+\left(<4..1\right)\cdot H+\left(<4..1\right)H_{2}O+{\mathrm{Si}}^{4+} \end{array}$$

Derived from the energy scheme in Figure 7d, the reaction of oxonium ions with the bulk silicon and the reaction with the hydrogen-terminated silicon are possible at the platinum–silicon contact. It is remarkable that this effect is almost exclusively due to the strong silicon valence band bending at the platinum–silicon contact (cf. Figure 3d). The underpotential effect of the 2H_3_O^+^/H_2_ half-cell at the platinum surface is the lowest of all metals. Nevertheless, the effect of the oxonium ions is limited precisely when the equilibrium potential of the 2H_3_O^+^/H_2_ half-cell of the solution falls below the level of 0 V vs. SHE.

The hypothetical limitation of the oxonium ion reduction to the threshold values mentioned was further verified experimentally. If the oxonium ions react with silicon at the metal–silicon contact, the reaction according to Equation (13) would have to produce hydroxyl radicals (∙OH) intermediately according to the scheme shown in Equation (14). The hydroxyl radicals are subsequently converted to H_2_O by the reaction with the intermediate hydrogen radicals.
(14)$$\mathrm{Si}+4{\;H}_{3}O^{+}\;\to {\;\mathrm{Si}}^{4+}+8\;\cdot H\;+\;4\;\cdot \mathrm{OH}\;⇌{\;\mathrm{Si}}^{4+}+2{\;H}_{2}+4{\;H}_{2}O$$

The intermediate formation of hydroxyl radicals is formally impossible in other conceivable reactions of silicon, for example, with H_2_O or HF. If oxonium ions were the reaction partner, the interception of intermediately formed hydrogen radicals should result in the formation of free hydroxyl radicals. Free hydroxyl radicals are able to react with all other species, including other silicon atoms, due to the very high standard potential of *E*^0^ = 2.26 ± 0.48 V vs. SHE [77]. It is assumed that the addition of a hydrogen radical scavenger would cause an increase in the amount of oxidized and dissolved silicon if oxonium ions were be involved in the reaction process.

Such a test was performed for gold and platinum deposition with and without the addition of para-Chlorobenzoic acid (pCBA), which acts as a hydrogen radical scavenger in an acidic solution [78]. This method is less suitable for silver and copper deposition because the less soluble [79] AgCl and CuCl compounds might form partially.

Figure 8 shows the results of the experiments in terms of silicon dissolution after *t* = 3600 s for gold deposition (*b Si (diss., t = 3600 s)*) and after *t* = 14,400 s for platinum deposition (*b Si (diss., t = 14,400 s)*) in relation to the initial metal ion molalities *b Me (diss., t = 0)*. The results for *b Si (diss., t = 3600s)* or *b Si (diss., t = 14,400 s)* without the addition of pCBA were taken from the results shown in Figure 2c,d and 6c,d.

The silicon dissolution was significantly increased in the presence of pCBA compared to tests without the addition of pCBA if the initial metal ion molalities were below the threshold values of *b Au (diss., t = 0)* = 1.5 × 10^−4^ mol∙kg^−1^ or *b Pt (diss., t = 0)* = 6.0 × 10^−5^ mol∙kg^−1^. Above these threshold values, the addition of pCBA had no effect compared with the tests without pCBA. These results confirm that oxonium ions are involved as an oxidizing agent below the threshold values. Above the threshold values, there must be another oxidizing agent.

Coincidence of the oxonium ion reduction thresholds with the significant decreases in copper, gold and platinum deposition rates is deducible (cf. Figure 2b–d). An explanation is the coupling of the reaction of the oxonium ion to the silicon with the silicon dissolution process. For this, the oxonium ion has to react as an ion pair with fluoride H_3_O^+^∙F^−^, as has been proposed by Giguère and Turell [80].

The oxonium ion oxidizes the silicon, and the fluoride binds to the oxidized silicon, destabilizing the neighboring Si–Si bonds and promoting further oxidation of the silicon either by further oxonium ion–fluoride pairs or the metal cations, as shown in Figure 9. If the oxonium ion reduction is inhibited, the process of silicon dissolution slows down, and thus, the continuation of the metal cation reduction does too. Above the threshold values, the process of silicon dissolution apparently changes toward the previously postulated anionic HF species F^−^, HF_2_^−^ and H_2_F_3_^−^.

#### 3.6.2. The Divalent and Tetravalent Reaction Mechanism (*Section II*) 

Within the *section II*, the divalent and tetravalent valence exchange between the metal cations and the bulk silicon is assumed, as has been proposed in the reaction schemes of Equations (1) and (3). There is a slight overlap of *section I* with *section II* when comparing the threshold values of *b Me (diss., t = 0)*.

*Section II* has its formal starting point where two electronic holes are transferred from the metal cation reduction to silicon. According to the charge numbers of the metal cations considered, these points correspond with a stoichiometric Δ*b Ag*:Δ*b Si* ratio of 2:1 mol∙kg^−1^:mol∙kg^−1^, a Δ*b Cu*:Δ*b Si* ratio of 1:1 mol∙kg^−1^:mol∙kg^−1^, a Δ*b Au*:Δ*b Si* ratio of 2:3 mol∙kg^−1^:mol∙kg^−1^ and a Δ*b Pt*:Δ*b Si* ratio of 1:2 mol∙kg^−1^:mol∙kg^−1^. The intersections with the functions *f*_1_ shown in Figure 6 are *b Ag (diss., t = 0)* = 1.3 × 10^−4^ mol∙kg^−1^, *b Cu (diss., t = 0)* = 8.1 × 10^−5^ mol∙kg^−1^, *b Au (diss., t = 0)* = 3.1 × 10^−5^ mol∙kg^−1^ and *b Pt (diss., t = 0)* = 5.1 × 10^−5^ mol∙kg^−1^. The gold and platinum deposition thresholds correspond to the first inflection points of the deposition rate *k*_95%_, shown in Figure 2c,d.

The tetravalent valence exchange occurs where the stoichiometric Δ*b Me*:Δ*b Si* ratios converge to Δ*b Ag*:Δ*b Si* = 4:1 mol∙kg^−1^:mol∙kg^−1^, Δ*b Cu*:Δ*b Si* = 2:1 mol∙kg^−1^:mol∙kg^−1^, Δ*b Au*:Δ*b Si* = 4:3 mol∙kg^−1^:mol∙kg^−1^ and Δ*b Pt*:Δ*b Si* = 1:1 mol∙kg^−1^:mol∙kg^−1^. These values correspond to the first maxima of the functions *f*_1_ (cf. Figure 6 and Table 1). The *f*_1_ functions in silver and copper deposition reach these maxima at initial metal ion molalities of *b Me (diss, t = 0)* ≈ 1.0 × 10^−2^ mol∙kg^−1^. However, in gold deposition, *section II* ends at *b Au (diss, t = 0)* = 1.5 × 10^−4^ mol∙kg^−1^ when the postulated Schottky barrier to the valence band of the bulk silicon becomes effective (cf. Figure 2c and Figure 3c). At this point, the maximum stoichiometric Δ*b Au*:Δ*b Si* ratio is 1:1 mol∙kg^−1^:mol∙kg^−1^ (cf. function *f*_1_ in Figure 6c). *Section II* is much narrower in platinum deposition for the same reason. The reaction with the bulk silicon ends at *b Pt (diss, t = 0)* = 6.0 × 10^−5^ mol∙kg^−1^ with a maximum stoichiometric Δ*b Pt*:Δ*b Si* ratio of 3:4 mol∙kg^−1^:mol∙kg^−1^ (cf. Figure 2d and Figure 6d).

The main difference in *section I* is that another oxidant has to fill the gap in the valence balance when fewer than four valences are exchanged between metal cation and silicon. The participation of the oxonium ions is excluded for the reaction process in *section II*, as discussed in *section I*. It is assumed that either H_2_O or a HF species has to be the secondary oxidant. In terms of the concentration, H_2_O with a molality between *b* = 54 and 55.5 mol∙kg^−1^ is more favored compared to *b HF (diss., t = 0)* = 0.06 mol∙kg^−1^ up to 1.5 mol∙kg^−1^. Another indicator for H_2_O is that there is no significant dependence of metal deposition kinetics on the HF molality in silver and copper deposition experiments without the delay in silicon dissolution. The experiments with the delay of the silicon dissolution also indicate that the metal deposition was limited, probably because of the intermediate formation of silanol groups or oxides which can only be formed by reactions with H_2_O (cf. Figure 2a,b).

The formulation of the reaction of H_2_O with silicon under the formation of SiO_2_, as found in the literature [18,28,38,41,43,81,82], does not appear to be suitable for metal deposition. The redox potential of this reaction *E*^0^ = −0.84 V vs. SHE [28,43] is too weak for the valence transfer via the valence band mechanism at the metal–silicon contact. It is more plausible that the silicon is first oxidized to the unstable species Si^2+^ [83] by the metal cation reduction and that only secondarily does H_2_O change Si^2+^ to Si^4+^ according to Equation (15), as has been previously postulated by Memming and Schwandt [53].
(15)$${\mathrm{Si}}^{2+}+2{\;H}_{2}O\;\to {\;\mathrm{Si}}^{4+}+2\cdot H\;+\;2{\;\mathrm{OH}}^{-}\;⇌{\;\mathrm{Si}}^{4+}+H_{2}+2{\;\mathrm{OH}}^{-}$$

After the formation of Si^4+^, intermediate silanol groups or oxides are converted to the final product SiF_6_^2−^ by the reaction with the anionic HF dissociation products F^−^, HF_2_^−^ and H_2_F_3_^−^ proposed. The entire reaction process in *section II* can be summarized with Equation (16).
(16)$$\begin{array}{cc} \left(\frac{2}{z}..\frac{4}{z}\right){\;\mathrm{Me}}^{z+}+ & \left(2..0\right){\;H}_{2}O+6{\;F}^{-}\left({\mathrm{and}\;\mathrm{HF}}_{2}{}^{-}{\mathrm{and}\;H}_{2}F_{3}{}^{-}\right)+{\mathrm{Si}}_{\mathrm{bulk}}\\ & ⟶\left(\frac{2}{z}..\frac{4}{z}\right)\mathrm{Me}+\left(2..0\right)\cdot H+\left(2..0\right){\mathrm{OH}}^{-}+{\mathrm{SiF}}_{6}{}^{2-} \end{array}$$

#### 3.6.3. The Change from Bulk-Si Oxidation to Si–H_x_ Oxidation (*Section III*)

If a certain threshold value of *b Me (diss., t = 0)* is exceeded, the Me^z+^/Me half-cells are able to oxidize the hydrogen-terminated silicon atoms via the valence band mechanism. As discussed for the kinetic evaluation of the experiments, the redox potentials of the Me^z+^/Me half-cell required depends on the energy level of the valence band of the hydrogen-terminated silicon at the metal–silicon contact. This parameter, in turn, depends on the difference in work functions between the metal and the hydrogen-terminated silicon Δ*E*, which is further influenced by the dipole moment at the metal-solution interface (cf. Figure 3).

Silver: Regarding silver deposition, this process is enabled at a redox level of the Ag^+^/Ag half-cell of 0.65 V vs. SHE (cf. Figure 2a and Figure 3a). This corresponds to an initial Ag^+^ molality of *b Ag (diss., t = 0)* = 5.5 × 10^−3^ mol∙kg^−1^ (cf. Figure 2a and Figure 6a). At this point, the maximum stoichiometric Δ*b Ag*:Δ*b Si* ratio is ≈ 4:1 mol∙kg^−1^:mol∙kg^−1^ (cf. *f*_1_ in Figure 6a). This finding can be explained by an equal oxidation of the Si–H, Si=H_2_ and Si≡H_3_ groups according to the reaction scheme in Figure 10a. The maximum stoichiometric Δ*b Ag*:Δ*b Si* ratio at higher initial Ag^+^ molalities converges towards a ratio of 5:1 mol∙kg^−1^:mol∙kg^−1^ (cf. *f*_1_ in Figure 6a).

This ratio can result if the Si=H_2_ and Si≡H_3_ groups but not the Si–H groups were involved in the oxidation process according to the reaction scheme in Figure 10b. This finding relates to the experiments for which a low activity of anionic fluoride species was postulated for the silicon dissolution process (cf. Figure 4a and Figure 6a). This leads to the conclusion that the Si–H terminations could not be dissolved sufficiently quickly after the initial oxidation and remained, for example, as silanol groups. These groups would be passivated against further oxidation by Ag^+^, so that only the Si=H_2_ and Si≡H_3_ groups participated in the further reaction course.

Copper: The oxidation of the hydrogen-terminated silicon by the Cu^2+^/Cu half-cell is theoretically impossible, because the redox potential of this half-cell is at least 0.1 eV below the level of the valence band of the hydrogen-terminated silicon at the copper–silicon contact (cf. Figure 2b and Figure 3b). Therefore, only the oxidation of the bulk silicon, according to the reaction scheme of Figure 10c, would be plausible (*section II*). However, the Δ*b Cu*:Δ*b Si* ratio in the experiments with initial Cu^2+^ molalities of *b Cu (diss, t = 0)* ≥ 8.2 × 10^−3^ mol∙kg^−1^ exceeded the stoichiometric ratio of 2:1 mol∙kg^−1^:mol∙kg^−1^ and converged to a maximum ratio of 3:1 mol∙kg^−1^:mol∙kg^−1^ (cf. *f*_1_ in Figure 6b). This result is possible under two conditions. The first requirement is that an accumulation of Cu(I) species must have occurred during the experiments. The second requirement is that only the Si≡H_3_ groups have been oxidatively attacked. The redox potential of the Cu^+^/Cu half-cell is higher than that of the Cu^2+^/Cu half-cell [62]. With a Cu^+^ molality of 8.2 × 10^−3^ mol∙kg^−1^, the Cu^+^/Cu half-cell reaches a level of 0.40 V vs. SHE. Hypothetically, this is the energetic level from which the valence transfer via the valence band of the hydrogen-terminated silicon is enabled. Based on the energy scheme shown in Figure 3b, it would be possible if the amount of band bending Δ*E* at the copper–silicon contact is 0.28 eV and not 0.21 eV, resulting in an energy level of the valence band of the hydrogen-terminated silicon at the copper–silicon contact of −0.40 eV. The sufficient accumulation of the intermediate Cu^+^ would be conceivable if the reduction from Cu^2+^ to metal state had been limited. Such a limitation was proven for the respective experiments due to delayed silicon dissolution (cf. Figure 4b). It is further assumed that especially the silicon atoms of the oxidized Si–H and Si=H_2_ groups were not dissolved sufficiently fast and Si−OH and Si=(OH)_2_ groups were formed. As a result, only the Si≡H_3_ groups remained freely accessible. The proposed reaction scheme based on these considerations is shown in Figure 10d.

Gold: Above an initial AuCl_4_^−^ molality of *b Au (diss., t = 0)* = 1.0 × 10^−2^ mol∙kg^−1^, the maximum Δ*b Au*:Δ*b Si* ratio in gold deposition converged towards a ratio of 4:3 mol∙kg^−1^:mol∙kg^−1^. From this finding, the reaction scheme shown in Figure 10e can be concluded, according to which the Si–H, Si=H_2_ and Si≡H_3_ groups are oxidized stoichiometrically uniformly. The process did not start at this ratio in the experiment with the highest initial AuCl_4_^−^ molality of *b Au* (*diss.*, *t* = 0) = 9.3 × 10^−2^ mol∙kg^−1^, but at Δ*b Au*:Δ*b Si* ≈ 1:3 mol∙kg^−1^:mol∙kg^−1^, and increased subsequently to the maximum ratio of 4:3 mol∙kg^−1^:mol∙kg^−1^.

This deviating finding can be explained by the fact that the silicon dissolution was limited under the conditions of this experiment. The *a (F^−^ + HF_2_^−^ + H_2_F_3_^−^) (t = 0)*:*b Si (diss., t_95%_)* ratio was below the critical level of 6:1 mol∙kg^−1^:mol∙kg^−1^ at ≈ 1:1 mol∙kg^−1^:mol∙kg^−1^ proposed. It is assumed that hydrogen-terminated silicon groups were not sufficiently available at the beginning of the experiment due to the retarded silicon dissolution and silanol group formation. Consequently, the valence transfer shifted to the Si–Si backbonds of the near-surface silicon atoms, according to the reaction scheme in Figure 10f. It is supposed that the activity of the anionic HF species increased with decreasing AuCl_4_^−^ molality. As a result, the silicon dissolution and re-hydrogen termination process became faster, so that the reaction could switch to the reaction scheme of Figure 10e.

Platinum: The maximum Δ*b Pt*:Δ*b Si* ratio of 6:5 mol∙kg^−1^:mol∙kg^−1^ shown in Figure 6d indicates that the Si–H groups were not oxidized, but the Si=H_2_ and Si≡H_3_ groups were, according to the reaction scheme in Figure 10g. Unlike the other metal depositions, this is not caused by delayed silicon dissolution. It is assumed that the valence transfer to the Si–H groups is inhibited by a Schottky barrier. According to the calculated findings of Papaconstantopoulos and Economou [84], the energy levels of the valence bands of the silicon with different hydrogen terminations vary. The less hydrogen bound to the silicon, the lower the energy level of the valence band of the silicon. Due to the high difference in work functions between platinum and hydrogen terminated silicon (cf. Figure 3d), the valence band of the silicon of the Si–H termination at the platinum–silicon contact was probably bent to just above the level of the Fermi energy, so that the Schottky barrier was constituted to this group but not to the Si=H_2_ and Si≡H_3_ groups.

Analogously to gold deposition, a Δ*b Pt*:Δ*b Si* ratio of ≈ 3:5 mol∙kg^−1^:mol∙kg^−1^ was determined for the platinum deposition experiment with the highest initial PtCl_6_^2−^ molality of *b Pt (diss., t = 0)* = 9.7 × 10^−3^ mol∙kg^−1^ at the beginning of the experiment, which later increased to the maximum ratio of 6:5 mol∙kg^−1^:mol∙kg^−1^ (cf. Figure 5d). This reaction process also indicates a temporarily slower silicon dissolution process. The *a (F^−^ + HF_2_^−^ + H_2_F_3_^−^) (t = 0)*:*b Si (diss., t_95%_)* ratio of 9:1 mol∙kg^−1^:mol∙kg^−1^ was just above the critical ratio of 6:1 mol∙kg^−1^:mol∙kg^−1^ proposed. In contrast to the gold deposition process sketched in Figure 10f, only a part of the oxidized silicon on the surface was temporarily not dissolved fast enough in the platinum deposition experiment. It is assumed, therefore, that some of the valences resulting from the PtCl_6_^2−^ reduction were transferred to the hydrogen-terminated silicon and the others to the Si-Si backbonds of the surface-near silicon. A mixed reaction can be interpreted from the initially proven Δ*b Pt*:Δ*b Si* ratio of about 3:5 mol∙kg^−1^:mol∙kg^−1^, according to the scheme of Figure 10h, which must have changed in the course of the experiment to the reaction scheme postulated in Figure 10g.

#### 3.6.4. The Decrease in Δb Me:Δb Si Ratios toward a Minimum (*Section IV*)

The reaction processes according to *section III* are followed by a successive decrease in the stoichiometric Δ*b Me*:Δ*b Si* ratios to the level of the *f*_2_ function for silver and to the level of the *f_3_* functions for the other metal deposits (cf. Figure 6). The Δ*b Ag*:Δ*b Si* ratio converges to a minimum of 10:3 mol∙kg^−1^:mol∙kg^−1^ (cf. *f*_2_ in Figure 6a, Table 1). The minima of the other Δ*b Me*:Δ*b Si* ratios decrease the higher the initial metal ion molality was and tended towards a minimum ratio of Δ*b Cu*:Δ*b Si* = 1:2 mol∙kg^−1^:mol∙kg^−1^, Δ*b Au*:Δ*b Si* = 1:3 mol∙kg^−1^:mol∙kg^−1^ and Δ*b Pt*:Δ*b Si* = 1:2 mol∙kg^−1^:mol∙kg^−1^ (cf. *f_3_* in Figure 6b–d, Table 1).

This finding can be interpreted as follows: The initial reactions of the metal cations according to *section III* lead to the oxidization of the upper Si–H, Si=H_2_ and Si≡H_3_ groups and their consecutive dissolution (cf. Figure 10). Through the involvement of H_2_O and the anionic HF species, the silicon underneath the previously removed silicon will be re-terminated with hydrogen. The process step of metal cation reduction and silicon oxidation then starts again. Through the multiple sequence of the process, the metal successively penetrates the surface of the silicon. The consecutive reaction by H_2_O is becoming increasingly important due to the successive decreasing molality of the metal cations. The change of the reaction course is metal-dependent as the different minimum ratios of Δ*b Me*:Δ*b Si* indicate.

Silver: The minimum Δ*b Ag*:Δ*b Si* ratio of 10:3 mol∙kg^−1^:mol∙kg^−1^ in silver deposition leads to the conclusion that by reducing 10 Ag^+^, formally one Si–H and two Si=H_2_ groups have to be oxidized according to the reaction scheme in Figure 11a. This means that the Si≡H_3_ groups attacked initially (Figure 10a) would not be further oxidized with decreasing Ag^+^ molality. The shift towards a selective oxidation of the different hydrogen-terminated silicon groups can be explained by the fact that the energetic level of the Ag^+^/Ag half-cell decreases with the successive consumption of Ag^+^. If the Ag^+^ molality falls below a certain level, the energetic level of the valence band of the silicon of the Si≡H_3_ group can obviously no longer be reached. Based on the experimental findings, the level of *E_v_ (Si≡H_3_)* should be about −0.70 eV (cf. Figure 3a). The energy level corresponds to that of the Ag^+^/Ag half-cell at an Ag^+^ activity of 4.5 × 10^−2^ mol∙kg^−1^.

Copper: The determined minimum of the ratio Δ*b Cu*:Δ*b Si* of 1:2 mol∙kg^−1^:mol∙kg^−1^ indicates a change from the oxidation of the Si≡H_3_ groups via Cu^+^ reduction (cf. Figure 10d) to the oxidation of the Si–H groups formed after the removal of the Si≡H_3_ groups (cf. Figure 11b).

Gold: The trend of gold deposition from a maximum Δ*b Au*:Δ*b Si* ratio of 4:3 mol∙kg^−1^:mol∙kg^−1^ toward a minimum Δ*b Au*:Δ*b Si* ratio of 1:3 mol∙kg^−1^:mol∙kg^−1^ suggests that after the initial attack on all hydrogen-terminated silicon groups, according to the scheme of Figure 10e, the reaction has shifted successively to the Si–H and Si=H_2_ groups (Δ*b Au*:Δ*b Si* = 1:1 mol∙kg^−1^:mol∙kg^−1^), and furthermore, exclusively to the Si–H groups (Δ*b Au*:Δ*b Si* = 2:3 mol∙kg^−1^:mol∙kg^−1^) with decreasing AuCl_4_^−^ molality. After the AuCl_4_^−^ molality has decreased to the level of 1.5 × 10^−4^ mol∙kg^−1^, the bulk silicon would also have been accessible to oxidation again. The proven minimum Δ*b Au*:Δ*b Si* ratio of 1:3 mol∙kg^−1^:mol∙kg^−1^ could then result from the reaction with the Si–H termination and the underlying bulk silicon, according to the reaction scheme outlined in Figure 11c. The reaction process assumed is associated with a high formation of molecular hydrogen compared to the other metal deposits. According to the optical impression, the gas formation during gold deposition was strongest compared to the other metal depositions.

Platinum: The process of platinum deposition is only partially similar to gold deposition. The oxidation of the Si=H_2_ and Si≡H_3_ groups is preferred in *section III* (cf. Figure 10g). As the platinum deposition penetrates the silicon, the Δ*b Pt*:Δ*b Si* ratio decreases to a minimum of 1:2 mol∙kg^−1^:mol∙kg^−1^. This leads to the conclusion that the reaction path changes towards the Si–H groups with decreasing PtCl_6_^2−^ molality, as shown in Figure 11d. Only twofold valence transfer formally occurs in the reduction of Pt(IV) via the intermediate Pt(II) state to the Pt state. This has the consequence that no molecular hydrogen can be produced whose formation requires onefold valence transfers. This conclusion corresponds to the observation of the experimental course of platinum deposition, after which no gas formation was observable.

## 4. Conclusions

Based on batch experiments, the reductions of Ag^+^, Cu^2+^, AuCl_4_^−^ and PtCl_6_^2−^ on multi-crystalline silicon in a HF matrix was investigated regarding kinetics and stoichiometry.

Metal deposition started slowly and almost linearly in time in all experiments. After exceeding a metal deposition of about 2.0 × 10^−6^ mol per cm^2^; silicon surface, the subsequent metal deposition resulted in an exponential decay. The kinetics of both phases differed by a factor of 2 to 3. This result corresponds to findings described in the literature [12,36,37]. The reason was the change from the initial metal nucleation to the three-dimensional growth of the metal layer on the silicon.

The kinetics of metal deposition correlate positively with the molalities of Ag^+^, Cu^2+^, AuCl_4_^−^ and PtCl_6_^2−^. The decisive criteria were the overlap of the energetic level of the redox potential of the Me^z+^/Me half-cells involved with the valence band of the bulk and hydrogen-terminated silicon at the metal–silicon contact and a possibly existing Schottky barrier behavior, as has been postulated previously in [20,21,22,23,24,33,46,48].

The metal deposition kinetics for silver increased logarithmically with higher initial Ag^+^ molality in the energetic overlap range of the Ag^+^/Ag half-cell with the valence band of the bulk silicon. The energetic level of the valence band of the hydrogen-terminated silicon with a redox level of *E (Ag^+^/Ag)* = 0.65 V vs. SHE (*b Ag (diss., t = 0)* = 5.5 × 10^−3^ mol∙kg^−1^) was presumably reached, and the kinetics of silver deposition correlated potentially with the initial Ag^+^ molality. The maximum metal deposition rate was found at the initial Ag^+^ molality of *b Ag (diss., t = 0)* of 1.5 × 10^−1^ mol∙kg^−1^.

The kinetics of gold deposition were mirror-inverted compared to the findings for silver deposition. The kinetics of gold deposition correlated potentially with the initial AuCl_4_^−^ molality up to an initial AuCl_4_^−^ molality of *b Au (diss., t = 0)* of 3.1 × 10^−5^ mol∙kg^−1^ and reached the maximum at this threshold. Above it, the kinetics of gold deposition decreased to a minimum at *b Au (diss., t = 0)* of 1.5 × 10^−4^ mol∙kg^−1^ and increased above logarithmically. It is suspected that the valence band of the bulk silicon at the threshold is bent above the energetic level of the Fermi energy. Thus, a Schottky barrier inhibited the reaction path with the bulk silicon. The valence transfer shifted above the threshold to the hydrogen-terminated silicon.

Based on the findings of silver and gold deposition kinetics, it can be deduced that the kinetics of the reactions with bulk and hydrogen-terminated silicon are similar. The proposed faster reaction with the silicon atoms of the bulk silicon compared to the hydrogen-terminated silicon atoms [18,24,26,29,41] cannot be confirmed according to the results of this study.

A partial Schottky barrier behavior could be derived for platinum deposition similar to gold deposition if an initial PtCl_6_^2−^ molality of *b Pt (diss., t = 0)* = 5.1 × 10^−5^ mol∙kg^−1^ was exceeded. The Schottky barrier affected the reaction with the bulk silicon and also the reaction with the Si–H_x_ groups. The kinetic inhibition of platinum deposition was more pronounced compared to gold deposition. This can be attributed to the stronger band bending effect and Schottky barrier height due to the comparatively greater work function of platinum [65,67,68,69].

The investigation of the kinetics of copper deposition revealed a kinetic inhibition of copper deposition in the case of limited silicon dissolution. The decrease in the kinetics of copper deposition occurred when the activities of the HF dissociation products F^−^, HF_2_^−^ and H_2_F_3_^−^ had decreased below a ratio of 6:1 mol∙kg^−1^:mol∙kg^−1^ in relation to the molality of silicon dissolution. The criterion of limitation found reflects the stoichiometric ratio of the final product SiF_6_^2−^. Another relationship between other HF species, as insinuated in the literature in different variants [13,33,36,39,43,53,54,55,56,57,58,59], could not be verified.

The stoichiometric ratios of metal deposition determined in relation to silicon dissolution Δ*b Me*:Δ*b Si* indicate a complex reaction process. The stoichiometric Δ*b Me*:Δ*b Si* ratios increased with rising metal cation molalities within a sigmoidal functional relationship. At the lowest level of *b Me (diss., t = 0)*, the Δ*b Me*:Δ*b Si* ratios tended to ratios shortly above 0:1 mol∙kg^−1^:mol∙kg^−1^, indicating the involvement of at least one further oxidizing agent.

The assumption of a metal cation reduction with coexisting hydrogen ion reduction via the silicon conduction band mechanism [16,38,39,40] was not confirmed. The results and calculations indicate an oxonium ion reduction with valence transfer via the valence band of silicon at the metal–silicon contact. In the case of a sufficiently low molality of the metal cation, an underpotential effect of the 2H_3_O+/H_2_ half-cell is formed on the metal surface. If the resulting equilibrium potential of the 2H_3_O^+^/H_2_ half-cell reaches at least the energy level of the valence band of the bulk silicon, and at most, the level of the Fermi energy at the metal–silicon contact, the valence transfer from the oxonium ion to the silicon can occur via the valence band mechanism. Therefore, the molality of *b Ag (diss., t = 0)* = 3.3 × 10^−4^ mol∙kg^−1^, *b Cu (diss., t = 0)* = 2.5 × 10^−4^ mol∙kg^−1^, *b Au (diss., t = 0)* = 1.5 × 10^−4^ mol∙kg^−1^ or *b Pt (diss., t = 0)* = 6.0 × 10^−5^ mol∙kg^−1^ must not be exceeded. Otherwise, the process of oxonium ion reduction at the metal–silicon contact via the valence band mechanism is impossible.

Above these thresholds, the oxidation of the silicon will be caused by metal cation and consecutive water reduction. The valence exchange between metal cation and silicon is divalent at low molalities. According to the charge numbers of the Ag(I), Cu(II), Au(III) and Pt(IV) species, the Δ*b Me*:Δ*b Si* ratios are Δ*b Ag*:Δ*b Si* = 2:1 mol∙kg^−1^:mol∙kg^−1^, Δ*b Cu*:Δ*b Si* = 1:1 mol∙kg^−1^:mol∙kg^−1^, Δ*b Au*:Δ*b Si* = 2:3 mol∙kg^−1^:mol∙kg^−1^ and Δ*b Pt*:Δ*b Si* = 1:2 mol∙kg^−1^:mol∙kg^−1^. The missing valences for the oxidation of the silicon up to the oxidation state of +IV are produced by the reduction of water. At higher molalities, the proportion of tetravalent valence transfer increases and the proportion of water in the reaction process decreases. The functional relationship between the initial molalities of Ag^+^, Cu^2+^, AuCl_4_^−^ and PtCl_6_^2−^ and stoichiometric Δ*b Me*:Δ*b Si* ratios is sigmoidal. Above metal cation molalities of ≈ 1.0 × 10^−2^ mol∙kg^−1^, the valence exchange is tetravalent. At this level, the Δ*b Me*:Δ*b Si* ratios reach their maxima. With the exception of gold deposition, the stoichiometric ratios are not twice as high as in the divalent reaction range. The maximum Δ*b Ag*:Δ*b Si* ratio was 5:1 mol∙kg^−1^:mol∙kg^−1^. The maximum Δ*b Cu*:Δ*b Si* ratio amounted to 3:1 mol∙kg^−1^:mol∙kg^−1^, and the maximum Δ*b Pt*:Δ*b Si* ratio reached a level of 6:5 mol∙kg^−1^:mol∙kg^−1^. The valence balance implies the involvement of the hydrogen of the Si–H_x_ groups as a further reducing agent, because the oxidation of silicon above an oxidation state of +IV is not plausible. The criterion that the redox level of the Me^z+^/Me half-cells has to be at least equal to the energetic levels of the valence bands of the Si–H_x_ species is achieved for the initial molalities of *b Ag (diss., t = 0)* ≥ 5.5 × 10^−3^ mol∙kg^−1^. Much lower initial metal ion molalities are required in the gold and platinum deposition process, mainly due to the strong silicon valence band bending at the metal–silicon contact. If the threshold values of *b Au (diss., t = 0)* = 3.1 × 10^−5^ mol∙kg^−1^ and *b Pt (diss., t = 0)* = 5.1 × 10^−5^ mol∙kg^−1^ are exceeded, the valence transfer occurs exclusively to the valence band of the hydrogen-terminated silicon because the valence transfer to the bulk silicon atoms is inhibited by the constitution of a Schottky barrier. In copper deposition, the energetic level of the valence band of Si–H_x_ groups at the copper–silicon contact of –0.40 eV can be achieved by the Cu^+^/Cu half-cell at *b Cu (diss., t = 0)* = 8.2 × 10^−3^ mol∙kg^−1^. This process became significant in the experiments with limited silicon dissolution. The valence balance which can be derived from the maximum stoichiometric ratio of Δ*b Cu*:Δ*b Si* = 3:1 mol∙kg^−1^:mol∙kg^−1^ indicates that the reduction of Cu^+^ to Cu must have proceeded mainly with the oxidation of the Si≡H_3_ groups (→ 6 h^+^). The Δ*b Ag*:Δ*b Si* ratio of 5:1 mol∙kg^−1^:mol∙kg^−1^ for silver suggests an equal oxidation of the Si=H_2_ (→ 4 h^+^) and Si≡H_3_ (→ 6 h^+^) groups. In both cases, the reaction path was forced by the limited dissolution of previously oxidized bulk silicon and oxidized silicon of the former Si–H and partially Si=H_2_ groups induced by the low activity of F^−^, HF_2_^−^ and H_2_F_3_^−^. The Δ*b Au*:Δ*b Si* ratio of 4:3 mol∙kg^−1^:mol∙kg^−1^ for gold is due to an equal oxidation of all Si–H_x_ groups (Si–H (→ 2 h^+^), Si=H_2_ (→ 4 h^+^) and Si≡H_3_ (→ 6 h^+^)). The Δ*b Pt*:Δ*b Si* ratio of 6:5 mol∙kg^−1^:mol∙kg^−1^ can be explained by the assumption that the oxidation of the Si–H groups is inhibited by the constitution of a Schottky barrier, and thus, only the valence exchange with the Si=H_2_ and Si≡H_3_ groups can occur.

## Figures and Tables

**Figure 1 nanomaterials-10-02545-f001:**
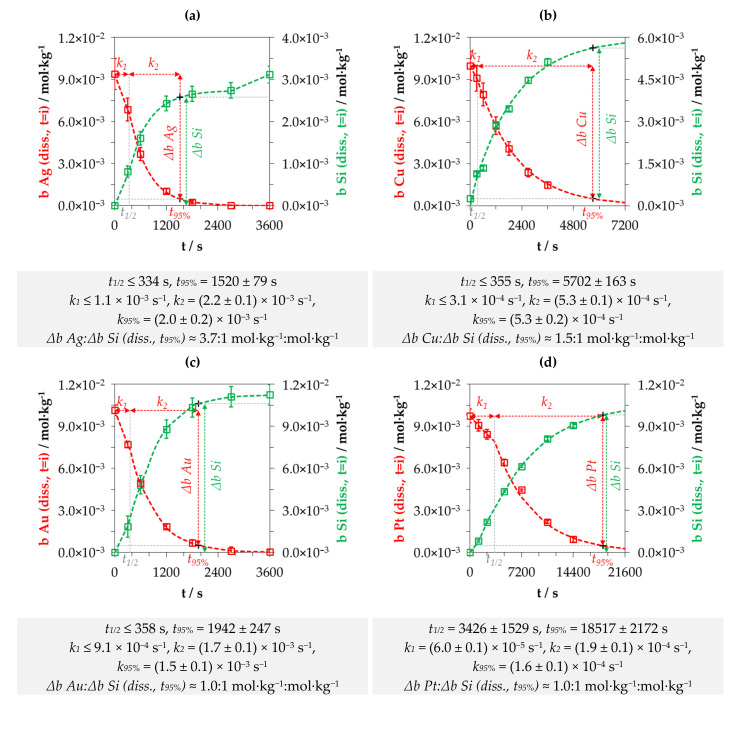
Dissolved (**a**) Ag, (**b**) Cu, (**c**) Au and (**d**) Pt species *b Me (diss., t = i)* and (**a**–**d**) Si species *b Si (diss., t = i)* at *t = i*; the period of the metal deposition phases 1 (*t* = 0 until *t*_1/2_) and 2 (*t*_1/2_ until *t*_95%_); the metal deposition rates *k*_1_ and *k*_2_; the *t*_95%_-weighted deposition rate *k*_95%_; and the stoichiometric ratio of the change in molality of the dissolved metal ion *Δb Me* in relation to the change in molality of the dissolved silicon species Δ*b Si* in the period of *t* = 0 until *t*_95%_ (Δ*b Me*:Δ*b Si (diss., t_95%_)*).

**Figure 2 nanomaterials-10-02545-f002:**
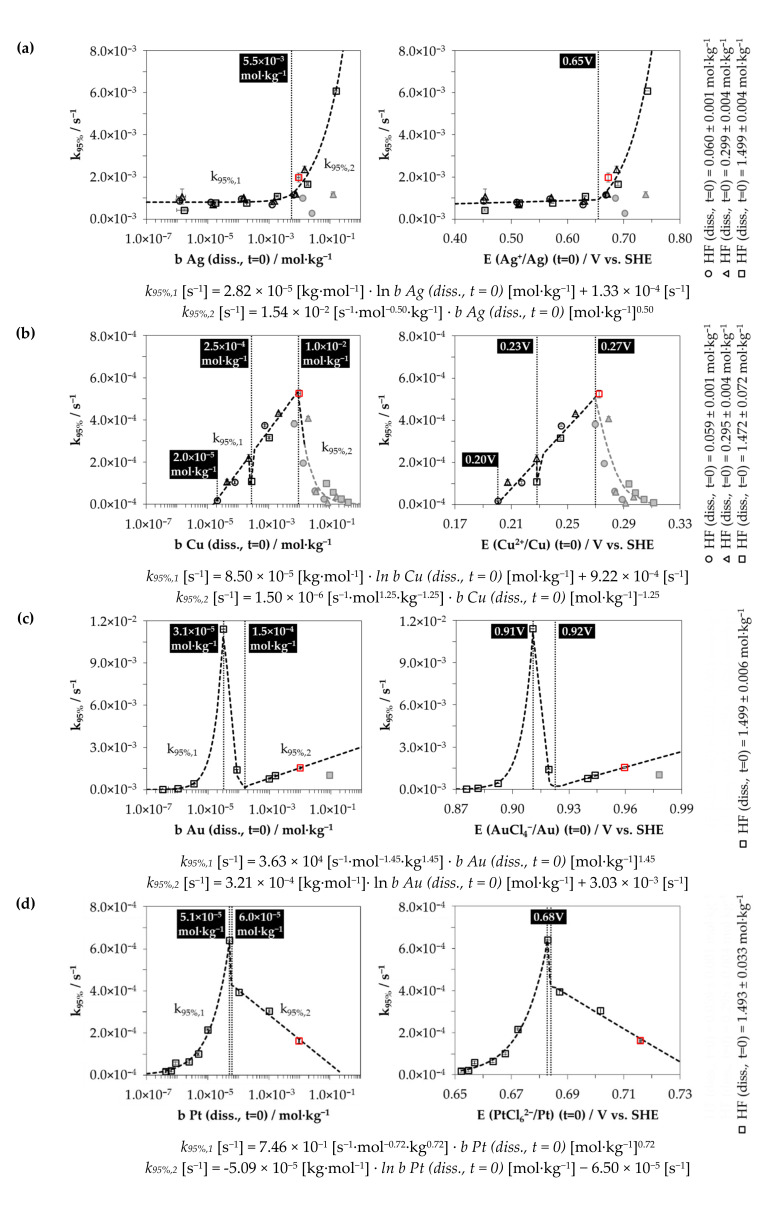
Metal deposition rates *k*_95%_ (*k*_95%,1_ and *k*_95%,2_) of the (**a**) Ag, (**b**) Cu, (**c**) Au and (**d**) Pt depositions in relation to the initial metal ion molality (*b Me (t = 0)*, Me = Ag, Cu, Au, Pt) or to the redox potential of the Me^z+^/Me half-cells *E (Me^z+^/Me)*. Red dots = examples from Figure 1, grey dots = findings with delayed silicon dissolution.

**Figure 3 nanomaterials-10-02545-f003:**
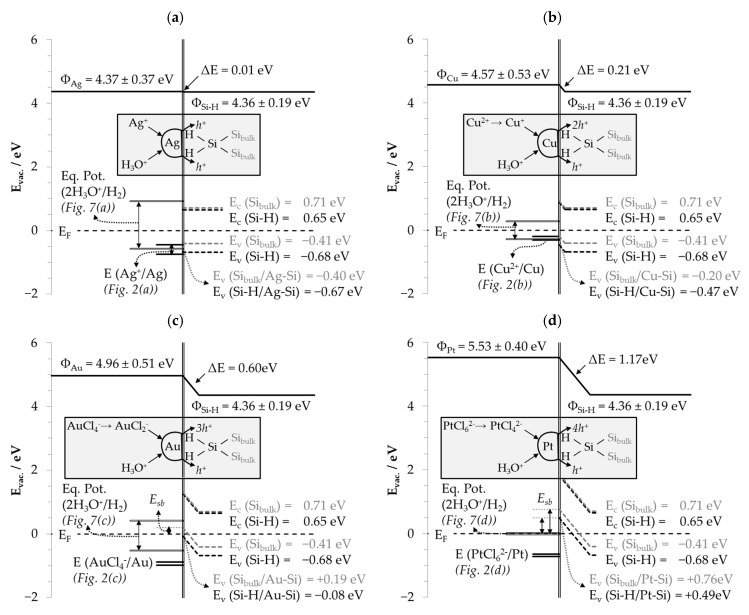
Work functions *Φ* of the metals (**a**) Ag, (**b**) Cu, (**c**) Au and (**d**) Pt and that of hydrogen-terminated Si in vacuum [65,66,67,68,69,70]; the difference in work functions Δ*E*; the Fermi energy level *E_F_* and that of the valence and conduction band of hydrogen-terminated silicon [71] (*E_v_ (Si–H) or E_c_ (Si–H)*) and of bulk silicon [72,73] (*E_V_ (Si_bulk_)* or *E_c_ (Si_bulk_)*); the energy levels of the valence bands at the metal–silicon contact (*E_v_ (Si_bulk_/Me-Si*) or *E_v_ (Si–H/Me-Si)*); the Schottky barrier *E_sb_*; the redox levels of the Me^z+^/Me half-cells *E (Me^z+^/Me)*; and the equilibrium potentials of the 2H_3_O^+^/H_2_ half-cells at the metal surface *Equation Pot. (2H_3_O^+^/H_2_)*.

**Figure 4 nanomaterials-10-02545-f004:**
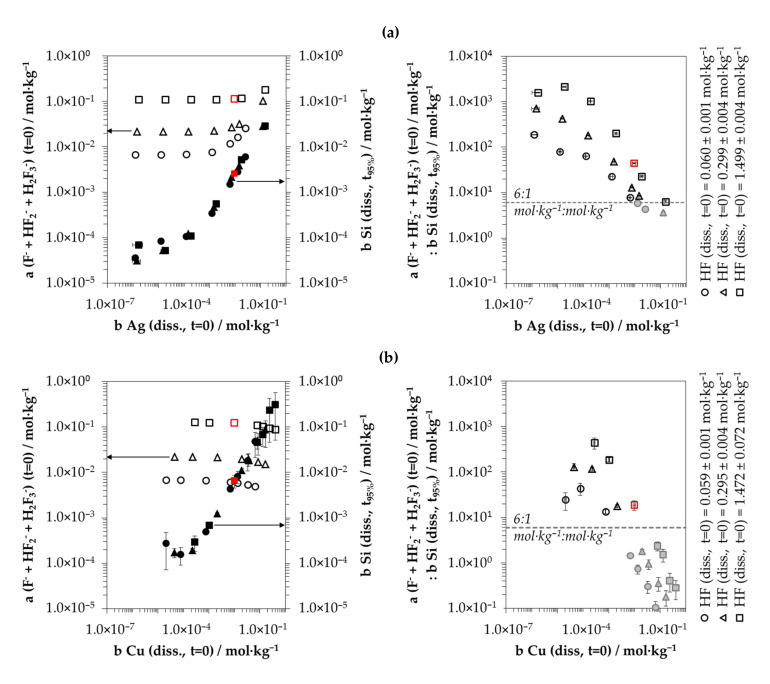
*Left:* Sum of the initial activities of the anionic HF dissociation products F^−^, HF_2_^−^ and H_2_F_3_^−^ (*a (F^−^ + HF_2_^−^ + H_2_F_3_^−^) (t = 0)*) and molality of the dissolved silicon at *t*_95%_ (*b Si (diss., t_95%_)*) in relation to the initial metal ion molality: (**a**) *b Ag (diss., t = 0)* or (**b**) *b Cu (diss., t = 0)*. *Right*: Quotient of *a (F^−^ + HF_2_^−^ + H_2_F_3_^−^) (t = 0)* and *b Si (diss., t_95%_)* in relation to *b Ag (diss., t = 0)* or *b Cu (diss., t = 0)*. (Red dots = examples from Figure 1, grey dots = findings with delayed silicon dissolution).

**Figure 5 nanomaterials-10-02545-f005:**
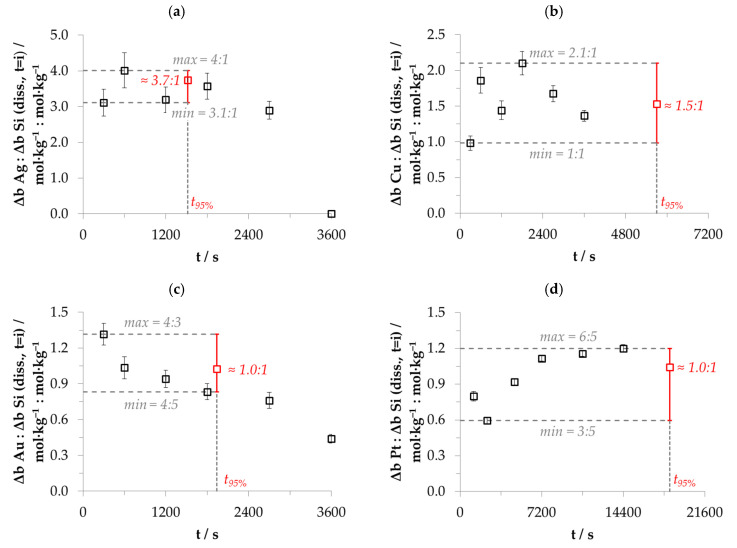
Stoichiometric ratios between the metal deposition Δ*b Me* ((**a**) Ag, (**b**) Cu, (**c**) Au and (**d**) Pt) and silicon dissolution Δ*b Si* as a function of time *t* (Δ*b Me*:Δ*b Si (diss., t = i)*) based on the experiments of Figure 1 with an indication of the *t*_95%_-weighted stoichiometric ratios (red dots) and the corresponding minima and maxima (bars) denoted as (Δ*b Me*:Δ*b Si (diss., t_95%_)*).

**Figure 6 nanomaterials-10-02545-f006:**
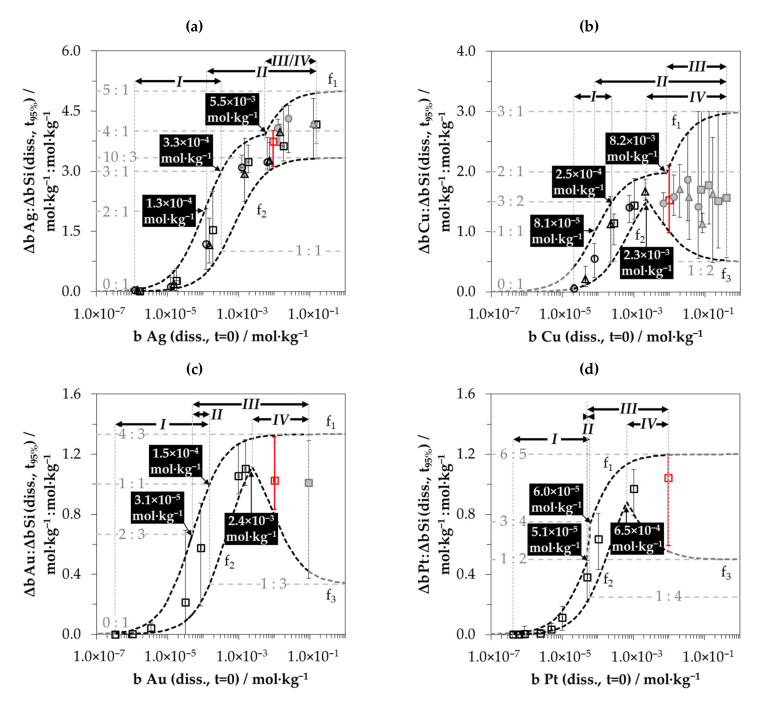
Stoichiometric ratios between the metal deposition Δ*b Me* ((**a**) Ag, (**b**) Cu, (**c**) Au and (**d**) Pt) and silicon dissolution Δ*b Si* until *t*_95%_ (Δ*b Me*:Δ*b Si (diss., t_95%_)*), in relation to the initial metal ion molality (*b Me (diss., t = 0)*), are indicated as points for the *t*_95%_-weighted values and as bars for the corresponding minima and maxima with delimitation by sigmoid functions *f*_1_, *f*_2_ and *f_3_* and classification of the underlying processes in *sections I*, *II*, *III* and *IV* (red dots = examples from Figure 1, grey dots = findings with limited silicon dissolution).

**Figure 7 nanomaterials-10-02545-f007:**
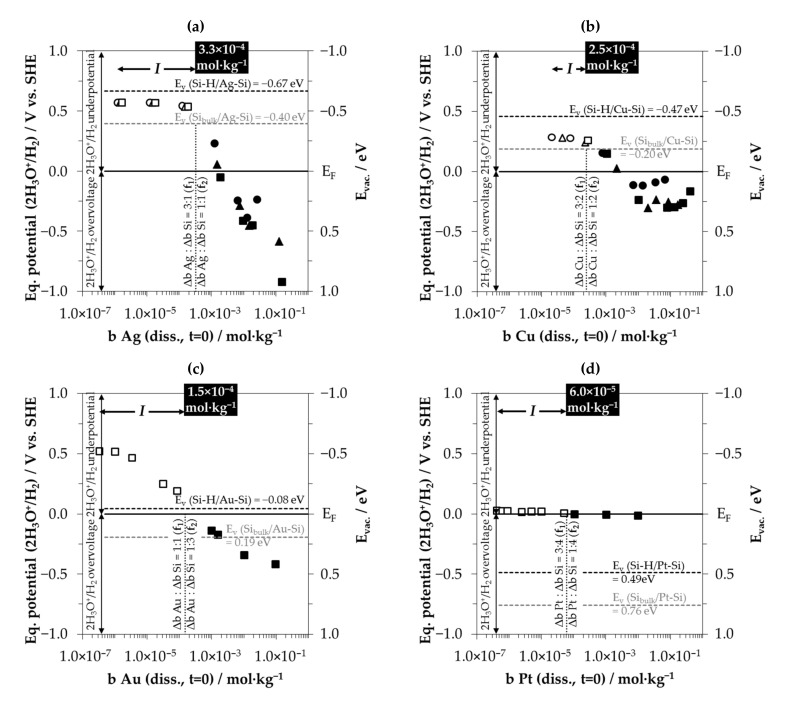
Equilibrium potentials of the 2H_3_O^+^/H_2_ half-cells at the (**a**) Ag, (**b**) Cu, (**c**) Au or (**d**) Pt surfaces (bright metal state/colloidal metal solution [76]) in relation to the initial metal ion molalities (*b Me (diss., t = 0)*), the level of the Fermi energy *E_F_* and levels of the valence bands of the bulk silicon (*E_v_ (Si_bulk_/Me-Si*) and that of the hydrogen-terminated silicon (*E_v_ (Si–H/Me-Si*)) at the metal–silicon contact. (Hollow symbols = oxonium ion reduction to molecular hydrogen possible, full symbols = no reaction).

**Figure 8 nanomaterials-10-02545-f008:**
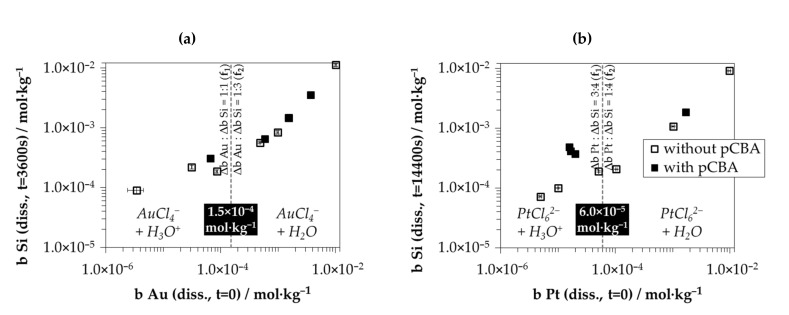
Molality of dissolved silicon in (**a**) gold deposition after *t* = 3600 s (*b Si (diss., t = 3600 s)*) and (**b**) platinum deposition after *t* = 14,400 s (*b Si (diss., t = 14,400 s*) in relation to the initial metal ion molality (*b Au (diss., t = 0)* and *b Pt (diss., t = 0)*, respectively) with and without the presence of para-chlorobenzoic acid (pCBA).

**Figure 9 nanomaterials-10-02545-f009:**
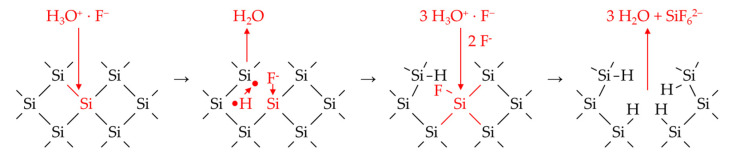
Proposed scheme of the reaction of the ion pair H_3_O^+^∙F^−^ with silicon.

**Figure 10 nanomaterials-10-02545-f010:**
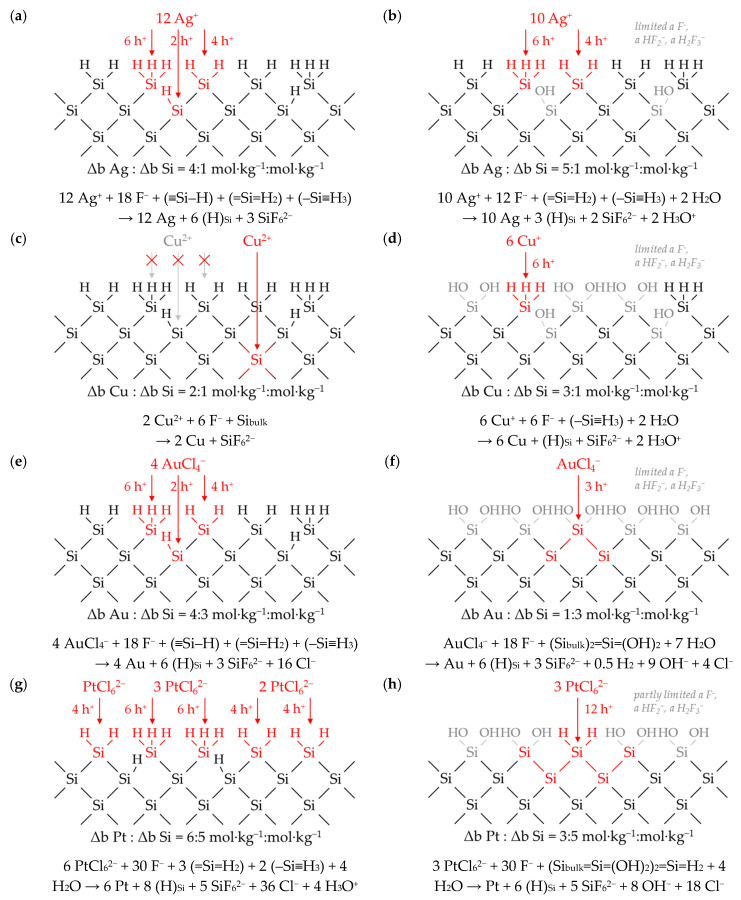
Reaction schemes of (**a**,**b**) silver, (**c**,**d**) copper, (**e**,**f**) gold and (**g**,**h**) platinum deposition on hydrogen-terminated silicon within *section III*.

**Figure 11 nanomaterials-10-02545-f011:**
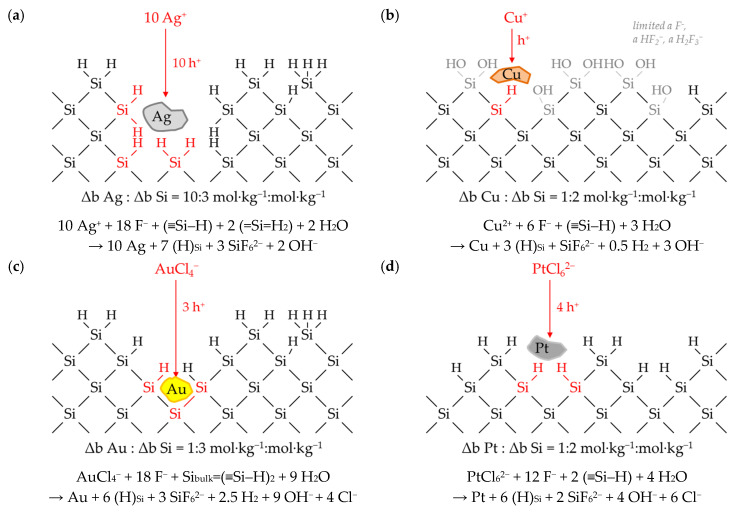
Reaction schemes of (**a**) silver, (**b**) copper, (**c**) gold and (**d**) platinum deposition on hydrogen-terminated silicon within *section IV*.

**Table 1 nanomaterials-10-02545-t001:** Parameterization of the sigmoidal functions *f*_1_, *f*_2_ and *f_3_* according to Equations (10) and (11).

	Parameter	Unit	Ag		Cu		Au	Pt	
*f* _1_	*b Me (diss., t = 0) valid range*	mol	7.1 × 10^−7^	5.5 × 10^−3^	2.1 × 10^−5^	8.2 × 10^−3^	3.3 × 10^−7^	4.2 × 10^−7^	6.0 × 10^−5^
		kg	5.5 × 10^−3^	1.3 × 10^−1^	8.2 × 10^−3^	4.1 × 10^−1^	9.3 × 10^−2^	5.1 × 10^−5^	9.7 × 10^−3^
	*(*Δ*b Me*:Δ*b Si)_max_*	mol∙kg^−1^	4:1	5:1	2:1	3:1	4:3	1:1	6:5
		mol∙kg^−1^							
	*x*	mol^0.99^	3.0 × 10^−5^	3.3 × 10^−4^	4.5 × 10^−5^	1.5 × 10^−3^	4.1 × 10^−5^	5.1 × 10^−5^	3.5 × 10^−5^
		kg^0.99^							
*f* _2_	*b Me (diss., t = 0) valid range*	mol	7.1 × 10^−7^–1.3 × 10^−1^		2.1 × 10^−5^–2.3 × 10^−3^		3.3 × 10^−7^	4.2 × 10^−7^–6.5 × 10^−4^	
		kg					2.4 × 10^−3^		
	*(*Δ*b Me*:Δ*b Si)_max_*	mol∙kg^−1^	10:3		2:1		4:3	6:5	
		mol∙kg^−1^							
	*x*	mol^0.99^	2.5 × 10^−4^		4.0 × 10^−4^		3.7 × 10^−4^	2.1 × 10^−4^	
		kg^0.99^							
*f_3_*	*b Me (diss., t = 0) valid range*	mol	---		2.3 × 10^−3^–4.1 × 10^−1^		2.4 × 10^−3^	6.5 × 10^−4^–9.7 × 10^−3^	
		kg					9.3 × 10^−2^		
	*(*Δ*b Me*:Δ*b Si)_min_*	mol∙kg^−1^	---		1:2		1:3	1:2	
		mol∙kg^−1^							
	*x*	mol^0.99^	---		3.5 × 10^−3^		9.0 × 10^−3^	1.2 × 10^−3^	
		kg^0.99^							

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
