# Peer review of "The Kinetics and Stoichiometry of Metal Cation Reduction on Multi-Crystalline Silicon in a Dilute Hydrofluoric Acid Matrix"

_nanomaterials, 2020, doi:10.3390/nano10122545_

Round 1

Reviewer 1 Report

The work by S. Schönekerl and J. Acker reports on a huge piece of work dedicated to reveal the kinetics of silver, gold, palladium and copper ions on silicon. I have no remarks on the scientific part. It is clear that the extensive experimental studies were performed and carefully analyzed. However, I have two major concerns:

1) The manuscript has to be restructured and reorganized (including figures). I understand that it is not easy to find the best way to present all these results, but it the current form it is very difficult to read which can significantly reduce the potential interest to the readers and impact of the work.

2) I do not feel Nanomaterials journal to be the best suited for this kind of research. The manuscript could be transferred to another journal within the same publisher (e.g. Processes or Materials).

Author Response

Dear reviewer,

Thank you for taking the time to review our extensive paper.

We are grateful for your valuable comments. According to your recommendation, the structure of the manuscript was revised in order to increase the clarity of the presented results. In particular, the chapter “results and discussion” was reorganized into several chapters to achieve a better understanding.

Finally, we would be grateful, if our manuscript would be considered for further processing in “nanomaterials”.

Kind regards

Stefan Schönekerl and Jörg Acker

Reviewer 2 Report

The authors present an impressive, very elaborate study of an important topic both for understanding the fundamentals of surface chemistry and electrochemistry, as well as for the semiconductor industry, i.e. the deposition of metals on silicon surface in HF solution, either in the context of 'desired' and 'undesired', spontaneous deposition. Experimental results and procedures are well described and the interpretation of results has been done in a clear and meticulous fashion. I can recommend publication of the work as is and have no further questions.   

Line 245 'functionally positively potentially' > This reads confusing and I assume a few words were left behind from text rewriting ? 

Author Response

Dear reviewer,

Thank you for taking the time to review our extensive paper.

We have corrected the phrase in line 245 that you noted.

Kind regards

Stefan Schönekerl and Jörg Acker

Reviewer 3 Report

As declared by the authors, the aim of this study was to determine and compare the kinetics of silver, copper, gold and platinum deposition on silicon, studying the influence of different concentrations of ions.

The experimental data are well presented and discussed and the conclusion are well supported by the final results.

For these reasons, in my opinion, the paper is suitable for publication

Author Response

Dear reviewer,

Thank you for taking the time to review our extensive paper.

Kind regards

Stefan Schönekerl and Jörg Acker